# Experience-Driven Exploration for Efficient API-Free AI Agents

## Abstract

Most existing software lacks accessible Application Programming Interfaces (APIs), requiring agents to operate solely through pixel-based Graphical User Interfaces (GUIs). In this API-free setting, large language model (LLM)-based agents face severe efficiency bottlenecks: limited to local visual experiences, they make myopic decisions and rely on inefficient trial-and-error, hindering both skill acquisition and long-term planning. To address these challenges, we propose **KG-Agent**, an experience-driven learning framework that structures an agent's raw pixel-level interactions into a persistent State-Action Knowledge Graph (SA-KG). KG-Agent overcomes inefficient exploration by linking functionally similar but visually distinct GUI states, forming a rich neighborhood of experience that enables the agent to generalize from a diverse set of historical strategies. To support long-horizon reasoning, we design a hybrid intrinsic reward mechanism based on the graph topology, combining a state value reward for exploiting known high-value pathways with a novelty reward that encourages targeted exploration. This approach decouples strategic planning from pure discovery, allowing the agent to effectively value setup actions with delayed gratification. We evaluate KG-Agent in two complex, open-ended GUI-based decision-making environments (Civilization V and Slay the Spire), demonstrating significant improvements in exploration efficiency and strategic depth over the state-of-the-art methods.

## 1 Introduction

The emergence of Large Language Models (LLMs) has accelerated a new generation of agentic AI systems Qin et al. (2025) capable of tackling complex tasks across a diverse range of domains, including web browsing Zheng et al. (2024), operating mobile and desktop applications Zhang et al. (2025b), crafting and exploration in virtual worlds Wang et al. (2023b), and even robotics scenarios Brohan et al. (2023). Initially, the dominant paradigm relied on agents based on predefined Application Programming Interfaces (APIs), where human designers decompose high-level goals into structured workflows and tools Yao et al. (2023); Shinn et al. (2023). While this approach ensures reliability on well-defined benchmarks Xie et al. (2024), its dependence on task-specific APIs fundamentally limits the agent's adaptability and scalability in open-ended environments. Bridging this gap, recent progress in Vision Language Models (VLMs) Zhou et al. (2022) has enabled Graphical User Interfaces (GUIs)-based agents Zhang et al. (2025a). These agents, such as UFO Zhang et al. (2024b) and CogAgent Hong et al. (2024), interact not only through APIs but also by observing and manipulating GUIs in a human-like manner. By integrating visual understanding with reasoning, they provide more general automated control and a richer, more intuitive mode of interaction. The pursuit of ultimate generality and human-like autonomy takes this progression a final step further, leading to the API-free GUI-based agents, as exemplified by CRADLE Tan et al. (2025) and Bottom-Up Agent Du et al. (2025). This methodology envisions agents that operate exclusively through a universal, human-style interface—using only screen pixels as input and keyboard/mouse actions as output. Such agents have demonstrated the remarkable ability to autonomously acquire skills from scratch in complex games and diverse software applications without any privileged access.

Despite these significant advances, as shown in Figure 1 (a), the API-free GUI-based agents face two fundamental challenges Du et al. (2025); Tan et al. (2025) that obstruct their path toward Artificial General Intelligence (AGI) Morris et al. (2024): (i) *Inefficient Exploration*: Without task-specific priors or APIs, skills must be discovered and validated purely through exhaustive trial-and-

(a) API-Free GUI-based Agent      (b) Experience-Driven Knowledge Framework

Figure 1: From Local Memory to Global Strategy. (a) A conventional API-free GUI-based agent relies on a simple skill database, where experiences are isolated. Its decision-making is a reactive trial-and-error process guided by myopic rewards like visual change, leading to inefficient exploration. (b) Our proposed KG-Agent leverages a structured knowledge graph as its skill library. This graph connects experiences, enabling a strategic "exploit or explore" mechanism, and the learning of API-free GUI-based agent is guided by an advanced implicit reward that captures long-term value, accelerating knowledge transfer and fostering strategic planning.

reasoning. This leads to a sample inefficiency so severe that agents typically require 2-2.5 times more environment interactions to match the progression of prior-assisted baselines. (ii) *Limited Long-Horizon Strategic Reasoning*: The prevailing reliance on myopic reward signals, e.g., visual changes, fails to incentivize the multi-step plans essential for sophisticated gameplay. This inability to value setup actions with delayed gratification severely limits strategic depth. Overcoming these challenges requires methods that accumulate reusable abstractions from experience and reward formulations that capture long-term value, thus enabling robust planning, transfer, and generalization across diverse tasks.

To overcome these limitations, we introduce **KG-Agent**, a novel experience-driven learning framework designed to transform an agent's pixel-based GUI interactions into structured, actionable knowledge. Central to our approach is a persistent, cross-episode State-Action Knowledge Graph (SA-KG) that serves as the agent's long-term memory. This graph organizes raw visual observations into state nodes and models acquired skills as edges between them, thereby converting unstructured pixel-level experience into a coherent network for strategic decision-making in API-free environments. To directly combat *inefficient exploration*, KG-Agent connects functionally analogous yet visually distinct GUI states through similarity edges, forming a rich *neighborhood of experience*. This enables the agent's reasoning module to query entire functional neighborhoods, generalizing from diverse historically successful strategies rather than relying on isolated, myopic data points. To address *limited long-horizon reasoning*, we leverage the graph topology to design a novel hybrid intrinsic reward mechanism. This system combines a *state value reward* (quantifying strategic potential based on outgoing connections) with a *novelty reward* for environmental discovery. This potential-based formulation is crucial for incentivizing setup actions that yield delayed gratification, enabling robust long-term planning in open-ended environments. Our contribution is three-fold:

- We propose **KG-Agent**, a framework that structures raw GUI interaction experience into a persistent SA-KG, introducing a *neighborhood of experience* to combat inefficient exploration by enabling generalization across functionally similar but visually distinct states.

- We design a hybrid reward mechanism derived from the SA-KG's topology that addresses limited long-horizon reasoning. By combining a potential-based *state value reward* for strategic exploitation with a *novelty reward* for targeted exploration, our approach effectively incentivizes multi-step plans with delayed gratification.

- We conduct extensive experiments in two complex, open-ended environments (Civilization V and Slay the Spire), demonstrating that **KG-Agent** significantly enhances exploration efficiency and strategic depth compared to state-of-the-art baselines.

## 2 RELATED WORK

**LLM-based Agents.** The application of LLM-based agents to complex, multi-step tasks has yielded remarkable achievements across domains Xi et al. (2025), including web browsing Gu et al. (2024), software operation Jin et al. (2024), and robotics Firoozi et al. (2025). Video games pose a particularly demanding testbed, requiring precise low-level control alongside high-level planning, abstraction, and adaptation Li et al. (2025b). Recent advances have shown strong performance in environments like Minecraft Li et al. (2025a) and StarCraft II Ma et al. (2024), yet early work often relied on privileged APIs for simplified observations and semantic action spaces Wang et al. (2023a), limiting applicability to closed-source commercial games that expose only raw pixels. With the rise of multi-modal capabilities, GUI-based agents have emerged Zhang et al. (2025a), such as OpenAI Operator, UFO Zhang et al. (2024b), and CogAgent Hong et al. (2024), which integrate visual understanding with text-based reasoning to enable more general and intuitive control of diverse applications.

**API-free Agents.** In pursuit of the ultimate, human-like General AI, a new paradigm of agents that operate purely from raw pixel inputs without any API support has emerged. Exemplified by works like CRADLE Tan et al. (2025) and the Bottom-Up Agent Du et al. (2025), these systems learn complex skills from scratch via a universal human-style interface (screen, keyboard, and mouse), using an autonomous trial-and-error loop to discover and refine behaviors. This API-free GUI-based approach has proven capable of achieving tangible progress in complex open-ended environments. However, this promising paradigm introduces a critical challenge: experience siloing. Agents often retrieve memory in a localized and myopic way—relying on the most visually similar past state—failing to generalize across functionally similar but visually different contexts. As a result, they fall back on inefficient trial-and-error relearning, hindering knowledge accumulation and long-term strategy formation.

## 3 METHODOLOGY

As shown in Figure 2, the proposed **KG-Agent** is composed of three core components: a universal Environment IO Interface for open-ended environment interaction, a Memory System to store and structure experience, and a VLM-based Reasoning Module that directs the agent's behavior. Following Bottom-Up Agent Du et al. (2025), we model the environment as a Partially Observable Markov Decision Process (POMDP) Spaan (2012) defined by $(\mathcal{X}, \mathcal{A}, \mathcal{T}, \mathcal{R})$, where $\mathcal{X}$ is the visual observation space, $\mathcal{A}$ is the set of atomic actions, $\mathcal{T}$ represents the unknown transition dynamics, and $\mathcal{R}$ is the implicit reward signal. The agent is built around a VLM $f_{VLM}$ that, when conditioned on a prompt $prompt$ and current context $c_i$, produces function-specific outputs: $f_{VLM}(prompt, c_i) \rightarrow y_i$. A skill $\sigma$ is defined as a sequence of atomic actions $\sigma = (a_1, \ldots, a_k)$, each paired with a semantic descriptor $d_\sigma$, a natural language summary of its intent, generated by the VLM. The skill library $\mathbb{S} = \{\sigma_1, \ldots, \sigma_n\}$ evolves over time through discovery, refinement, and composition.

### 3.1 THE ENVIRONMENT IO INTERFACE

To operate in API-free environments, our agent requires a universal interface for perception and action. The Environment IO Interface, inspired by Bottom-Up Agent Du et al. (2025), fulfills this role, serving as the sole conduit through which the agent perceives and interacts with its environment. It operates exclusively on raw visual input and produces low-level, human-like actions, without access to internal states or privileged APIs. This design ensures our agent's generalizability while posing significant challenges in visual understanding and precise action execution.

**Environment Input: Purely Visual Observations.** The agent's perception is grounded exclusively in raw visual observations. At each time step $i$, the agent receives a screenshot $X_i \in \mathcal{X}$, which constitutes its entire sensory input. From this screenshot, a feature vector $x_i$ is extracted using the image encoder of a pre-trained CLIP model Radford et al. (2021) to represent the state. Notably, our agent forgoes the use of any Optical Character Recognition (OCR) models to extract textual content. This forces the agent to rely entirely on the VLM's intrinsic spatial perception and visual understanding to interpret the scene, without access to structured information.

**Environment Output: Simulated Mouse and Keyboard Operations.** The agent interacts with the environment by generating low-level keyboard and mouse commands from the action space $\mathcal{A}$,

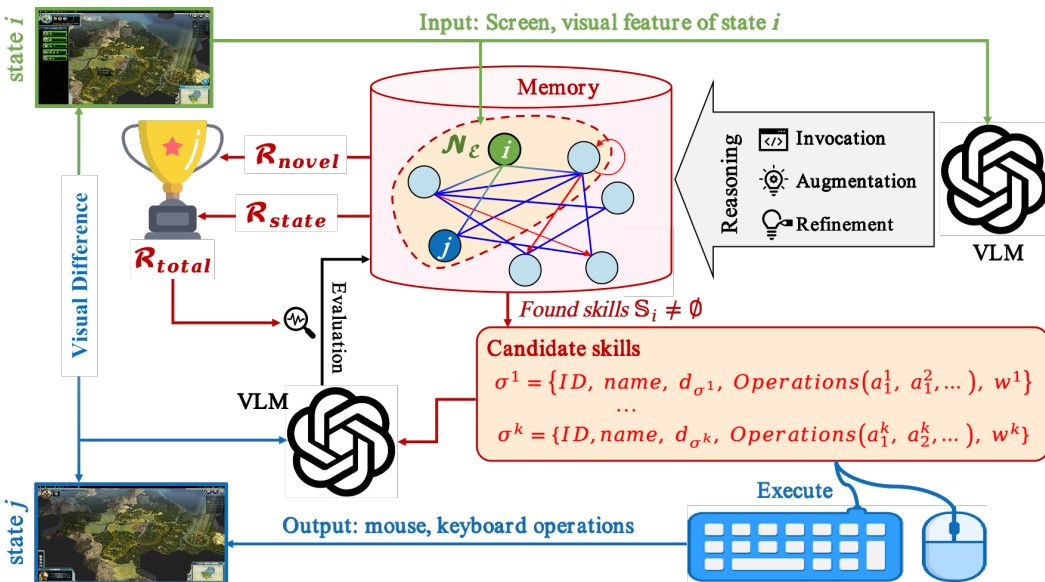

Figure 2: Overview of the proposed KG-Agent. For a given state $i$, the agent queries its memory to find a neighborhood of experience ($\mathcal{N}_\mathcal{E}$), where state nodes are connected by similarity edges (blue line) or skill edges (red arrow). If candidate skills $\mathbb{S}_i$ are retrieved, they are executed to transition the environment to state $j$. In addition, a VLM is central to this agent's reasoning, performing skill invocation, refinement, augmentation, and evaluation, where the refinement is guided by a hybrid implicit reward composed of *state value reward* and *novelty reward*.

mirroring human interaction. The action space includes all possible operations such as *click*, *drag*, *scroll*, and *type*. These can be combined in various ways to form combos and shortcuts. Considering that mouse operations can theoretically target any pixel on the screen, making the action space excessively large, we employ the Segment Anything Model (SAM) Kirillov et al. (2023) to dynamically identify and segment interactable UI elements from the current observation $X_i$. The interactable UI elements are also updated in the memory. A skill $\sigma$ is typically composed of one or more actions parameterized by the objects of interaction. For example, the skill 'Choose Production and Build Monument' translates to the operations: {'operate': 'Click', 'object_id': 11, 'object_name': 'Choose_Production'}; {'operate': 'Click', 'object_id': 24, 'object_name': 'Monument'}. To execute such an operation, the agent first retrieves a reference image of the target object (e.g., the 'Choose_Production' button) from memory. It then performs template matching against the current screen $X_i$ to find the object's precise location, calculating the center coordinates for the mouse click. This process grounds the VLM's symbolic action plan into concrete, executable operating system-level keyboard and mouse operations on the GUI.

### 3.2 EXPERIENCE-DRIVEN MEMORY SYSTEM

The memory of KG-Agent is designed to store and manage all useful information, enabling the agent to learn from past experiences and make informed decisions. It consists of two complementary components: a Procedural Memory and an SA-KG.

**Procedural Memory.** Inspired by Bottom-Up Agent Du et al. (2025), this component maintains the agent's immediate, operational knowledge, including executable skills and cached plans. It is implemented as a set of structured tables: i) *Objects Table*: Stores information about interactable UI elements identified by SAM, including their names and reference images. ii) *Skill Table*: Contains the evolving skill library $\mathbb{S}$, where each skill $\sigma$ is defined by its name, semantic descriptor $d_\sigma$, a sequence of operations $\{a_1, a_2, ..., a_k\}$, and a fitness score $\phi_\sigma$ that reflects its historical effectiveness. iii) *Action Clusters Table*: Groups skills that are applicable in similar states (identified by their feature vectors), facilitating efficient retrieval of contextually relevant skills. iv) *Monte Carlo Search Trees (MCST) Table*: Stores serialized search trees Browne et al. (2012) associated with specific states, allowing the agent to cache and reuse complex decision-making plans.

**State-Action Knowledge Graph (SA-KG).** The SA-KG is the cornerstone of the agent's long-term memory, designed specifically to overcome the bottleneck of inefficient exploration. It is a graph $\mathcal{G} = (\mathcal{V}, \mathcal{E})$ that transforms the agent's memory from a passive store of episodes into a semantically connected network that facilitates strategic reasoning and generalization. The SA-KG is constructed from nodes $\mathcal{V}$ and two types of edges: similarity edges $\mathcal{E}^{Sim}$ and skill edges $\mathcal{E}^{\sigma}$: i) *Nodes ($\mathcal{V}$)*: Each node $v_i \in \mathcal{V}$ represents a unique state, characterized by its CLIP feature vector $x_i$. A new observation with feature vector $x_j$ is merged into an existing node $v_i$ if their cosine similarity exceeds a threshold, i.e., $\cos(x_i, x_j) > \theta_{merge}$; otherwise, a new node $v_j$ is created. ii) *Similarity Edges ($\mathcal{E}^{Sim}$)*: An undirected edge $e_{i,j}^{\text{sim}}$ connects two functionally analogous but visually distinct states $v_i$ and $v_j$ if $\theta_{merge} > \cos(x_i, x_j) > \theta_{simi}$. Each edge has a weight $w_{i,j}^{\text{sim}} \in [0, 1]$ equal to the cosine similarity between $x_i$ and $x_j$. iii) *Skill Edges ($\mathcal{E}^{\sigma}$)*: A directed edge $e_{i,j}^{\sigma}$ from state $v_i$ to $v_j$ represents the successful execution of a skill $\sigma$. Each skill edge has a weight $w_{i,j}^{\sigma}$ quantifying the utility of that skill $\sigma$ for the specific state transition. Crucially, some actions produce a large immediate visual change but lead to a dead end, while a crucial setup move might have minimal impact but be key to a long-term strategy. To capture this, we define the skill edge weight $w_{i,j}^{\sigma}$ as a combination of immediate visual change and historical effectiveness:

$$w_{i,j}^{\sigma} = f_{sigmoid} \left( \alpha \Delta_{i,j} + (1 - \alpha) \frac{\phi_{\sigma}}{\phi_{\sigma} + C_0} \right), \tag{1}$$

where $f_{sigmoid}(\cdot)$ is the Sigmoid activation function, the $\Delta_{i,j}$ is the visual change ratio between states $v_i$ and $v_j$, the $\phi_{\sigma}$ is the historical fitness of skill $\sigma$, the $C_0$ is a constant that controls the sensitivity of the fitness term, and the $\alpha$ is a weighting factor that balances the importance of immediate change versus long-term strategic value. We quantify visual change as the proportion of changed pixels between consecutive grayscale screenshots. Pixels with an absolute difference exceeding 30 (0–255 scale) are counted, and the ratio $\Delta_{i,j}$ of such pixels to total pixels is computed. This design allows the agent to develop sophisticated, long-horizon plans by valuing both types of actions.

### 3.3 REASONING AND DECISION-MAKING MODULE

The reasoning module is the decision-making engine of KG-Agent, responsible for a continuous cycle of skill invocation, augmentation, refinement, and evaluation. It orchestrates a hierarchical decision-making process that leverages the structured knowledge in both the SA-KG and Procedural Memory to intelligently balance exploiting known strategies with exploring new possibilities. This entire process is guided by a novel, graph-based hybrid reward mechanism.

**Skill Invocation.** The agent employs a hierarchical, two-stage process for skill invocation that prioritizes structured, experience-based knowledge from the SA-KG before falling back to a more general VLM-guided search, ensuring both efficiency and reliability. Upon entering a new state $v_i$, the agent first identifies its *Neighborhood of Experience* $\mathcal{N}_{\mathcal{E}}(v_i)$, defined as the set of all nodes connected to $v_i$ by similarity edges:

$$\mathcal{N}_{\mathcal{E}}(v_i) = \{v_j | e_{i,j}^{sim}\}. \tag{2}$$

From this *Neighborhood of Experience* $\mathcal{N}_{\mathcal{E}}(v_i)$, it gathers a set of high-quality candidate skills $\mathbb{S}_{HQ}(v_i)$ by collecting all skills $\{\sigma_k\}$ found on outgoing skill edges from every node:

$$\mathbb{S}_{HQ}(v_i) = \bigcup_{v_j \in \mathcal{N}_{\mathcal{E}}(v_i)} \{\sigma_k \mid \exists v_l \text{ such that } e_{j,l}^{\sigma_k}\}, \tag{3}$$

where $e_{j,l}^{\sigma_k}$ represents a skill edge from a neighboring node $v_j$ to another node $v_l$ that is associated with skill $\sigma_k$. A probability distribution is then constructed to select a skill for execution. The probability $P(\sigma_k|v_i)$ of selecting a specific skill $\sigma_k$ from the candidate set $\mathbb{S}_{HQ}(v_i)$ is directly proportional to its corresponding weight, which we denote as $w^{\sigma_k}$. To form a valid probability distribution, this is normalized by the sum of all weights for all high-quality candidate skills:

$$P(\sigma_k|v_i) = \frac{w^{\sigma_k}}{\sum_{w^{\sigma_l} \in \mathbb{S}_{HQ}(v_i)} w^{\sigma_l}}. \tag{4}$$

From the distribution $P(\sigma_k|v_i)$, the agent samples up to $M$ skills for execution attempts. If this primary strategy fails to yield any successful skills, the agent performs a VLM-guided skill invocation from its Procedural Memory. This process queries the *Action Clusters Table* to retrieve a set of

candidate skills relevant to the current context. Formally, this is represented as:

$$\mathbb{S}_C(v_i) = \{\sigma_k \mid \sigma_k \in \mathbb{S} \ \wedge \ f_{VLM}(prompt^{invoke}, X_i, \mathbb{S}) = \sigma_k\}, \tag{5}$$

where $\mathbb{S}_C(v_i)$ denotes the candidate skills retrieved by the VLM based on the visual observation $X_i$. To avoid committing greedily in a stochastic and partially observable environment, the agent evaluates the candidate skills $\mathbb{S}_C(v_i)$ using an approach inspired by the Upper Confidence bound for Trees (UCT) algorithm Couëtoux et al. (2011), a core component of MCTS. For each candidate skill $\sigma_k \in \mathbb{S}_C(v_i)$, the agent calculates a potential utility score which balances exploitation of known, high-value skills with the exploration of less-tried options. This Upper Confidence Bound (UCB) score $\eta_{\sigma_k}$ is defined as:

$$\eta_{\sigma_k} = \phi(\sigma_k) + C_1\sqrt{\frac{\ln N_i}{n_k}} - P(\sigma_k), \tag{6}$$

where $\phi(\sigma_k)$ quantifies the fitness of skill $\sigma_k$. Initially set to 0, its value is updated based on two factors: detectable visual state transitions following execution, and the consistency of the skill's semantic description with the outcomes observed across before-and-after frames. The second term is the exploration bonus, where the $n_k$ is the execution count for skill $\sigma_k$, the $N_i$ is the total number of selections $\mathbb{S}_C(v_i)$, and the $C$ is a constant controlling the exploration-exploitation trade-off. $P(\sigma_k)$ is a penalty term that dynamically reduces the utility of skills whose prerequisite objects are not currently available, promoting the selection of fully completable actions. Finally, instead of deterministically picking the skill with the highest utility, the agent converts these scores into a probability distribution using a temperature-scaled softmax function. The probability of selecting skill $\sigma_k$ is given by:

$$P'(\sigma_k|v_i) = \frac{\exp(\eta_{\sigma_k}/\tau)}{\sum_{\sigma_l \in \mathbb{S}_C(v_i)} \exp(\eta_{\sigma_l}/\tau)}. \tag{7}$$

The skill to be executed is then stochastically sampled from this distribution. The temperature parameter $\tau$ dynamically adjusts the randomness of the selection. This method ensures a robust balance between exploiting proven strategies and exploring new possibilities. If the candidate set $\mathbb{S}_C(v_i)$ is empty, the agent defaults to skill augmentation in the current context.

**Skill Augmentation & Refinement.** In open-ended environments, where predefined APIs and priors are absent, most atomic actions $a \in \mathcal{A}$ are inherently task-irrelevant and semantically ambiguous. Consequently, discovering useful skills $\sigma = (a_1, \ldots, a_k)$ necessitates a structured trial-and-reasoning process, in which the agent explores diverse action combinations and evaluates their outcomes to identify meaningful behaviors. To manage the intractably large action space, we adopt the strategy by leveraging SAM to identify and segment UI elements, as well as constructing skills through incremental increases in sequence length $k$. Inspired by Bottom-Up Agent Du et al. (2025), our skill augmentation approach is driven by successful atomic action creation and validation rather than exhaustive combinatorial search. We partition the atomic action set $\mathcal{A}$ into three ordered subsets, i.e., $\mathcal{A}_1$, $\mathcal{A}_2$, and $\mathcal{A}_3$, to optimize the search space and accelerate exploration. The agent begins with $\mathcal{A}_1$, which contains previously validated single-step skills, then progresses to $\mathcal{A}_2$, comprising actions on recognized UI objects that prioritize semantically meaningful interactions, and finally considers $\mathcal{A}_3$, which includes residual actions such as clicks on unrecognized or background regions. The skill construction process starts with single-step actions ($k = 1$) and incrementally expands to longer sequences ($k = 2, 3, \ldots$). Each candidate skill $\sigma_k$ is formed by appending a new atomic action $a_k$ to a validated shorter skill $\sigma_{k-1}$. Skill expansion terminates as soon as an action sequence produces any recognizable and valuable effect, e.g., GUI transitions, measurable task progress, or meaningful state changes in the SA-KG—at which point the skill is annotated functionally and stored in memory.

To maintain an efficient and scalable skill library, we implement a dynamic skill pruning mechanism. MCTS is employed to ensure all skills receive sufficient testing opportunities. During skill execution, the agent records trajectory data and computes UCB scores according to Eq. 6. Skills that exceed the average visitation count and consistently demonstrate the lowest UCB scores are pruned from memory. This iterative process of expansion and pruning ensures that the skill library retains only high-utility behaviors, enabling complex, adaptive strategies to emerge compositionally from primitive actions. Furthermore, to maintain a compact and non-redundant skill set, the agent periodically triggers the VLM to consolidate the skill repertoire by clustering and merging functionally

Table 1: Performance comparison across two open-ended game environments, where Progression denotes in-game advancement, measured by floors cleared in *Slay the Spire* and turns survived in *Civilization V*; In-Game Scores are official run score in *Slay the Spire* and number of technologies unlocked in *Civilization V*; Execution-Responsive Rate is the percentage of predicted actions that lead to valid state transitions; and Token Costs ($) are average LLM tokens consumed per 100 steps, converted to USD for fair comparison across methods with varying episode lengths. Methods with * indicate prior-assisted setting, while those without * indicate zero-prior. NA denotes not applicable.

| Method | Slay the Spire | | | | Civilization V | | | |
|---|---|---|---|---|---|---|---|---|
| | Progression↑ (Floors) | In-game↑ Scores | Execution↑ Respons. Rate | Token↓ Costs | Progression↑ (Turns) | Techs↑ Researched | Execution↑ Respons. Rate | Token↓ Costs |
| GPT-4o* | $8.33 \pm 0.58$ | $48.73 \pm 2.43$ | $0.49 \pm 0.02$ | $1.01 \pm 0.03$ | $14.67 \pm 2.08$ | $1.00 \pm 0.00$ | $0.57 \pm 0.01$ | $0.91 \pm 0.02$ |
| Claude3.7* | $1.00 \pm 0.00$ | $5.00 \pm 0.00$ | $0.79 \pm 0.01$ | $1.22 \pm 0.04$ | $19.33 \pm 2.52$ | $3.33 \pm 0.58$ | $0.92 \pm 0.01$ | $1.18 \pm 0.09$ |
| UITARS-1.5* | $1.00 \pm 0.00$ | $5.00 \pm 0.00$ | $0.54 \pm 0.01$ | $0.49 \pm 0.24$ | $11.33 \pm 1.15$ | $1.33 \pm 0.58$ | $0.92 \pm 0.02$ | $0.12 \pm 0.01$ |
| CRADLE* | $1.00 \pm 0.00$ | $5.00 \pm 0.00$ | NA | $4.89 \pm 0.23$ | $0.00 \pm 0.00$ | $0.00 \pm 0.00$ | NA | $4.23 \pm 0.28$ |
| GPT-4o | $1.00 \pm 0.00$ | $5.00 \pm 0.00$ | $0.72 \pm 0.03$ | $1.27 \pm 0.09$ | $6.33 \pm 0.58$ | $0.00 \pm 0.00$ | $0.37 \pm 0.04$ | $0.76 \pm 0.03$ |
| Claude3.7 | $1.00 \pm 0.00$ | $5.00 \pm 0.00$ | $0.28 \pm 0.01$ | $1.08 \pm 0.06$ | $0.00 \pm 0.00$ | $0.00 \pm 0.00$ | $0.15 \pm 0.3$ | $1.22 \pm 0.13$ |
| UITARS-1.5 | $1.00 \pm 0.00$ | $5.00 \pm 0.00$ | $0.82 \pm 0.01$ | $\mathbf{0.09 \pm 0.01}$ | $0.00 \pm 0.00$ | $0.00 \pm 0.00$ | $0.63 \pm 0.01$ | $\mathbf{0.10 \pm 0.01}$ |
| BottomUp | $14.33 \pm 1.15$ | $91.33 \pm 9.61$ | $\mathbf{0.98 \pm 0.01}$ | $2.68 \pm 0.29$ | $59.67 \pm 8.74$ | $8.66 \pm 0.58$ | $0.92 \pm 0.01$ | $3.55 \pm 0.52$ |
| **KG-Agent** | $\mathbf{15.67 \pm 0.47}$ | $\mathbf{104.00 \pm 5.89}$ | $\mathbf{0.98 \pm 0.01}$ | $2.15 \pm 0.04$ | $\mathbf{107.33 \pm 6.55}$ | $\mathbf{13.33 \pm 1.25}$ | $\mathbf{0.93 \pm 0.01}$ | $3.03 \pm 0.17$ |

equivalent skills. This process is defined as:

$$\sigma' = f_{VLM}(prompt^{refine}, (X_i, \sigma, \mathcal{T})). \tag{8}$$

The dual process, i.e., combining population-based pruning with LLM-guided skill consolidation, ensures that $\mathbb{S}$ undergoes evolution towards a more reusable and semantically coherent state by eliminating redundancy and promoting general-purpose skills.

**Skill Evaluation.** For any skill $\sigma = (a_1, \ldots, a_k)$ that transitions the agent from state $v_i$ to $v_j$, the agent receives a resulting trajectory $\mathcal{T}$ from which it derives a behavioral signal. Since there is no external reward, the quality of the skill is implicitly evaluated via $\mathcal{R}_{total}$:

$$\mathcal{R}_{total} = \mathcal{R}_{progress} + \mathcal{R}_{semantic} + \mathcal{R}_{state} + \mathcal{R}_{novel}, \tag{9}$$

where $\mathcal{R}_{progress}$ evaluates the progress of the game via VLM to encourage the skill to lead to specific progress in the game, $\mathcal{R}_{semantic}$ measures the consistency via VLM between the predicted high-level effects of the skill and the actual results observed in the environment, $\mathcal{R}_{state}$ encourages the agent to execute skills that lead to states with greater future potential, $\mathcal{R}_{novel}$ serves as an intrinsic curiosity drive, directly rewarding the agent to explore and expand its knowledge.

To compute $\mathcal{R}_{state}$, we define a static skill potential $\mathcal{V}_S(v_i)$ as the sum of the weights of all outgoing skill edges. Then, the state-value reward for a transition from $v_i$ to $v_j$ is defined as the improvement in the agent's estimated long-term potential:

$$\mathcal{R}_{state} = \mathcal{V}_S(v_j) - \mathcal{V}_S(v_i) = \sum_{e \in \mathcal{E}^\sigma(v_j)} w^\sigma(e) - \sum_{e \in \mathcal{E}^\sigma(v_i)} w^\sigma(e). \tag{10}$$

A positive reward encourages movements to states from which higher future returns are expected. On the other hand, the novelty reward is a binary intrinsic incentive based on the prior discovery of a state. Let $\mathcal{V}_G$ be the set of all state vertices in the KG. The reward is defined as:

$$\mathcal{R}_{novel}(v_j) = \begin{cases} 1.000, & \text{if } v_j \notin \mathcal{V}_G \quad \text{(new state)}, \\ 0.015, & \text{if } v_j \in \mathcal{V}_G \quad \text{(known state)}. \end{cases} \tag{11}$$

This mechanism ensures a strong push to expand the boundaries of the known graph while providing a small, constant reward for re-visiting known states, preventing the agent from becoming completely stagnant in already-explored areas.

## 4 EXPERIMENTS

**Experimental Setup.** We adopt the same environments and evaluation metrics as the Bottom-up Agent Du et al. (2025) to facilitate direct performance comparison and clearly demonstrate the superior effectiveness of KG-Agent in exploration and strategic planning under identical API-free,

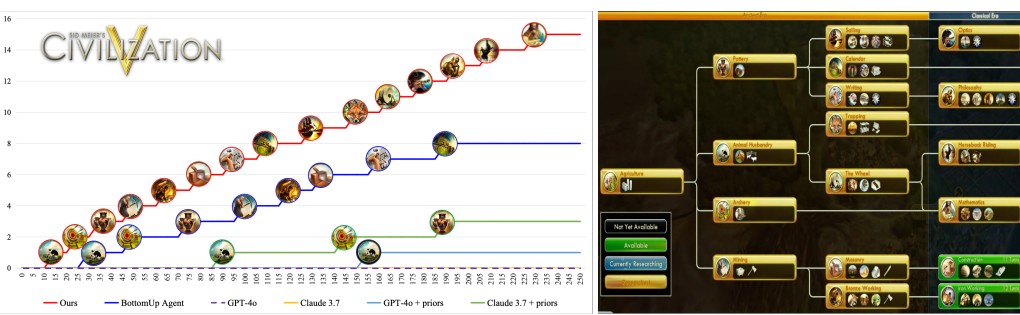

(a) Number of Technologies Researched      (b) Civilization V's Technology Tree

Figure 3: Game progression measured by Civilization V's technology tree. Our framework outperforms all baselines, including those with task-related priors.

Table 2: Evolution of skill library and SA-KG, as well as game performance over training rounds in *Civilization V*. Each round builds on previous memory with 100 steps.

| Round | Skill Library Information | | | SA-KG Information | | | Civilization V | | | |
|---|---|---|---|---|---|---|---|---|---|---|
| | Library Size | Skills Augment. | Skills Pruned | Nodes Size | Skill Edges | Simi. Edges | Progress.↑ (Turns) | Techs↑ Research. | Execution↑ Respons. Rate | Token↓ Costs ($) |
| Round 0 | 76 | 77 | 1 | 26 | 37 | 226 | 65 | 10 | 0.89 | 3.0 |
| Round 1 | 106 | 33 | 3 | 50 | 84 | 782 | 98 | 12 | 0.92 | 2.8 |
| Round 2 | 111 | 7 | 2 | 53 | 90 | 856 | 113 | 13 | 0.93 | 2.7 |
| Round 3 | 114 | 4 | 1 | 55 | 90 | 838 | 115 | 13 | 0.94 | 2.7 |

zero-prior, GUI-based conditions. Both *Slay the Spire* and *Civilization V* games require not only low-level execution accuracy but also high-level long-horizon planning, making them ideal testbeds for assessing strategic reasoning and adaptation. Their turn-based nature also allows seamless integration with LLM-based reasoning under current latency constraints. Due to the limitations of existing frameworks, which often rely on predefined APIs, task-specific prompts, or are ineffective even with prior knowledge (e.g., subgoals or game rules), we compare against several representative baselines: GPT-4o, Claude 3.7, the open-source model UI-TARS-1.5 Qin et al. (2025), the CRADLE Tan et al. (2025), and the Bottom-up Agent Du et al. (2025). As for CRADLE, we adapt its four core prompts for both games. Since CRADLE relies on predefined atomic skills, which both games do not provide, we convert KG-Agent's learned skills into CRADLE-compatible formats to ensure fair comparison. We use GPT-4o for semantic understanding and reasoning, and CLIP ViT-B/32 as the visual encoder. All hyperparameters remain consistent across experiments. For the SA-KG, the state merging threshold $\theta_{\text{merge}}$ is set to 0.95, and the similarity edge threshold $\theta_{\text{simi}}$ to 0.88. For skill edge weight computation (Eq. 1), we use a balancing factor $\alpha = 0.7$ and fitness sensitivity constant $C_0 = 5.0$. The UCT-based skill selection (Eq. 6) uses an exploration constant $C_1 = 5.0$. It is worth noting that *Slay the Spire* and *Civilization V* share the same agent implementation, including the underlying framework, hyperparameter settings, and prompt design. This consistency highlights the generality and adaptability of our API-free framework across diverse environments. Each agent is evaluated over three episodes per environment, with each episode ending upon game completion, failure, or after 500 steps. We report the mean and standard deviation of performance across these three runs to reflect both effectiveness and consistency.

## 4.1 COMPARATIVE RESULTS

As shown in Table 1 and Figure 3, the KG-Agent demonstrates statistically significant and consistent superiority over all baseline methods across two distinct open-ended game environments. Our agent achieves the highest progression metrics and maximum in-game scores, while maintaining exceptional execution reliability. This performance consistency across different game genres highlights the robustness of our approach. The contrast with CRADLE is particularly illuminating. While CRADLE relies on extensive prior knowledge encoded in hand-designed prompts and atomic skills, it achieves only minimal progression in both games. KG-Agent's substantial advantage without

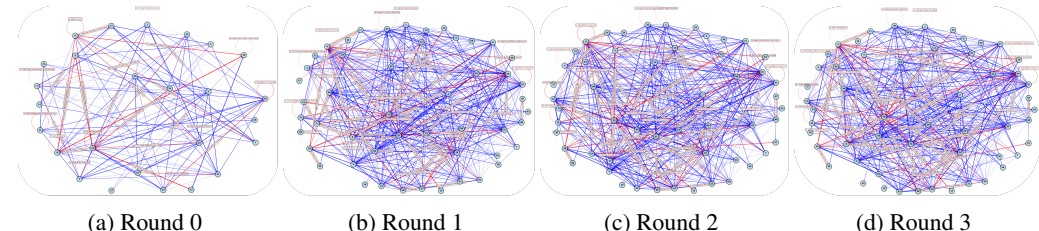

| (a) Round 0 | (b) Round 1 | (c) Round 2 | (d) Round 3 |

Figure 4: Visual evolution of the SA-KG from Round 0 to Round 3. Skill associations (red) and similarity links (blue) illustrate structural refinement and relational reasoning across rounds.

Table 3: Ablation study of core components of KG-Agent in *Civilization V*, with each run constrained to 100 steps for clarity. The "w/o similarity edge" means that each state forms an isolated node, with no merging or similarity-based connections permitted.

| Setting | Library Size | Node Size | Skill Edge | Simi. Edge | Turns↑ Survived | Techs↑ unlock | Execution↑ Respons. Rate | Token↓ Costs |
|---|---|---|---|---|---|---|---|---|
| **Full KG-Agent Model** | 76 | 26 | 37 | 226 | **65** | **10** | **0.89** | 3.0 |
| w/o similarity edge | 30 | 672 | 62 | 0 | 15 | 1 | 0.64 | 3.5 |
| w/o $\mathcal{R}_{novel}$ | 48 | 11 | 19 | 74 | 23 | 2 | 0.81 | 3.1 |
| w/o $\mathcal{R}_{state}$ | 57 | 20 | 33 | 110 | 39 | 4 | 0.87 | 3.2 |
| w/o $\mathcal{R}_{novel}$ & $\mathcal{R}_{state}$ | 64 | 18 | 39 | 96 | 35 | 3 | 0.79 | 3.3 |
| w/o $\mathcal{R}_{progress}$ & $\mathcal{R}_{semantic}$ | 33 | 9 | 14 | 42 | 14 | 1 | 0.59 | **2.1** |

any game-specific prior knowledge validates the effectiveness of our zero-prior learning paradigm. Furthermore, our method demonstrates remarkable consistency—achieving top performance across all metrics in both games, unlike baselines that show volatile performance between different environments. Notably, KG-Agent accomplishes this superior performance with reduced computational overhead, evidenced by lower token costs per 100 steps compared to the BottomUp Agent. These results confirm that our agent not only inherits BottomUp's execution reliability but significantly enhances strategic decision-making through our skill-aligned knowledge graph. The consistent excellence across all metrics underscores KG-Agent's effectiveness in mastering complex, open-ended environments through autonomous learning and adaptation.

## 4.2 ABLATION STUDIES

**Skill Evolution over Rounds**. Table 2 and Figure 4 illustrate the evolution of the skill library and SA-KG over four training rounds in *Civilization V*. The skill library grows dynamically through iterative augmentation and pruning, reflecting continuous refinement. The SA-KG also expands structurally, with similarity edges increasing notably in early rounds, highlighting enhanced relational reasoning, yet decreasing slightly by Round 3, indicating semantic consolidation and redundancy removal through functional merging. Concurrently, the number of nodes grows from 26 to 55 and skill edges from 37 to 90, with both effectively plateauing after Round 2, demonstrating that the graph converges toward a compact and functionally stable representation. These structural improvements correlate with steady gains in game progression and execution responsiveness, while token costs decrease, underscoring improved reasoning efficiency. The observed convergence across all metrics by Round 3 confirms that repeated skill reuse and structural consolidation facilitate sustained agent adaptation and strategic exploration in open-ended environments.

**Effect of Core Modules.** As shown in Table 3, removing similarity edges severely impairs long-term progression, while disabling the reward mechanism reduces strategic coherence. The performance drop in the "w/o similarity edge" condition stems from a deeper architectural inconsistency: although state merging and similarity edges were disabled, the graph-based rewards $\mathcal{R}_{novel}$ and $\mathcal{R}_{state}$ were retained. This created a mismatch: $\mathcal{R}_{novel}$ continuously rewarded the agent for registering every new state as a distinct node—promoting graph expansion rather than semantic exploration—while $\mathcal{R}_{state}$ operated on a fragmented graph with isolated nodes, causing unstable and localized value estimates. Consequently, the VLM was misled by pathological reward signals, pri-

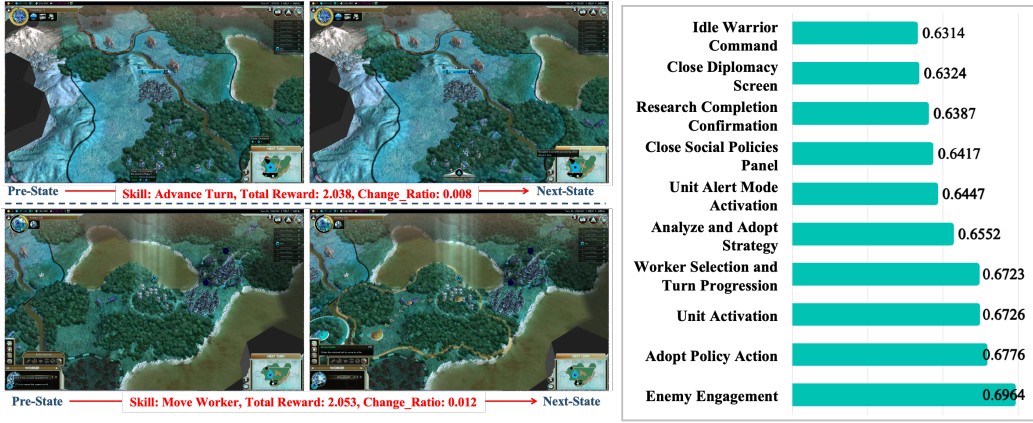

(a) Invocation with Low Visual Change but High Reward      (b) Skill Edges: Top 10 by Weight

Figure 5: Skill Invocation and critical skills. (a) Our framework executes high-reward skills even with minimal visual change between states. (b) The top 10 skill edges reflect a consistent focus on critical long-term actions, aligning with the game's core progression mechanics.

oritizing node proliferation over meaningful progress, as reflected in the exploded node count. The ablation of $\mathcal{R}_{progress}$ and $\mathcal{R}_{semantic}$ diminishes the total reward for skill evaluation, adversely affecting skill fitness $\phi(\sigma_k)$ and the subsequent UCB score in MCTS, thereby impairing skill selection and pruning. This leads to markedly poor performance across skill library size, connectivity, and in-game progress. These results underscore that our framework enables consistent state abstraction and stable value estimation, allowing graph-based rewards to deliver reliable learning signals for long-horizon planning. As shown in Figure 5a, our framework demonstrates its ability to prioritize strategic value over perceptual salience by invoking high-reward skills such as "Advance Turn" and "Move Worker" even when visual changes between states are minimal. This validates the reward mechanism's capacity to identify critical actions beyond superficial cues. Meanwhile, the key skills identified by the SA-KG, quantified in Figure 5b, reveal a consistent strategic focus. Skills with the highest edge weights, e.g., "Adopt Policy Action" (0.6776) and those related to unit and worker management, are precisely those that drive long-term progression, reflecting the critical role of the experience neighborhood in maintaining behavioral consistency and a coherent long-term strategy.

## 5    CONCLUSION

In this paper, we propose **KG-Agent** to organize pixel-level GUI interactions into a knowledge graph of states and actions, enabling agents to overcome short-sightedness in API-free, open-ended environments. By linking functionally similar yet visually distinct states, the agent generalizes past experiences into coherent long-term strategies. Experiments in open-ended environments demonstrate that KG-Agent achieves significant gains in exploration efficiency and strategic depth over state-of-the-art baselines. These results highlight the potential of structuring experience into a graph-based memory, advancing API-free agents toward scalable and general-purpose autonomy.

**Limitations and Future Work.** A key limitation of KG-Agent is its environment-specific knowledge, which lacks the abstract, transferable quality of human reasoning. Although it structures experience into a graph-based memory, the framework remains limited in generalizing across unseen tasks and diverse settings. Future work will focus on three directions: 1) enabling higher-level knowledge induction and reasoning across heterogeneous environments, 2) developing stronger graph pruning, state summarization, and hierarchical abstraction techniques to prevent structural bloat and support efficient cross-environment reasoning, and 3) building a more end-to-end GUI agent that uses natively capable VLMs for direct action grounding and a streamlined architecture to reduce complexity and improve generalization. These advances will help API-free agents approach human-like adaptability and robust general intelligence.

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

# A APPENDIX

## A.1 MORE RELATED WORKS

**LLM-Based Multi-Agent Systems.** The integration of LLMs Achiam et al. (2023); Liu et al. (2024a); Bai et al. (2023) into multi-agent systems has enabled a new architectural paradigm in which agents actively reason, communicate, and interact with external tools, environments, and users through structured interfaces Sun et al. (2024); He et al. (2025); Li et al. (2024). By leveraging APIs and GUI-based interactions, these systems elevate LLMs from passive text generators to dynamic participants capable of orchestrating complex, multi-step tasks Zhang et al. (2024a); Tzachristas (2024); Liu et al. (2024b). In practice, modern LLM-based agents function as autonomous or semi-autonomous entities that combine natural language understanding with tool-augmented reasoning Zhao et al. (2025); Wang et al. (2025). Moreover, GUIs further extend an agent's interactive capacity Tang et al. (2025a). In simulated or game environments, GUI perception modules can translate visual elements, e.g., buttons, maps, menus, into semantic representations that an LLM can reason about Hu et al. (2025). This approach enables LLM agents to operate across diverse digital environments, from software applications to strategy games, without requiring task-specific policy training Tang et al. (2025b); Liu et al. (2025). Notably, frameworks such as AutoGen Wu et al. (2024), MetaAgent Li et al. (2023), and Generative Agents Park et al. (2023) formalize these patterns, offering abstractions for agent roles, tool integration, message routing, and environment interfacing. By unifying natural language communication with API-driven and GUI-mediated interaction, LLM-based multi-agent systems are advancing from conceptual models to deployable platforms for open-ended, human-aligned reasoning and decision-making. Recently, the application of LLM-based multi-agents to complex, multi-step tasks has yielded remarkable achievements across diverse domains Xi et al. (2025), e.g., web browsing Gu et al. (2024), software operation Jin et al. (2024), and robotics Firoozi et al. (2025).

**API-Free LLM-Based AI Agents.** While early LLM-based agents frequently depended on privileged access to internal game states via APIs, this API-centric paradigm faces critical constraints Wang et al. (2023a). Its reliance on engineered interfaces inherently limits generalization, particularly in closed-source commercial games and proprietary software where internal APIs are inaccessible or undocumented Tan et al. (2025). Moreover, such abstractions can distance the agent from the rich visual context and fine-grained control characteristic of genuine human-computer interaction. In response, a more general and challenging direction has emerged: API-free LLM-based agents. These systems operate purely from raw pixel input, interacting through universal human-style interfaces such as keyboard and mouse, without relying on internal state APIs or specialized function calls Tan et al. (2025); Du et al. (2025). This approach targets ultimate generality, enabling agents to operate across any visible software environment, much like a human user, without custom integration. Notable progress has been made under this paradigm. In desktop automation, multimodal agents such as UFO Zhang et al. (2024b) and CogAgent Hong et al. (2024) interpret GUI elements and execute multi-step workflows based on natural language instructions, marking a stride toward generalist assistive agents. In open-ended game environments, systems like CRADLE Tan et al. (2025) and the Bottom-Up Agent Du et al. (2025) have demonstrated that LLM-based agents can learn complex tasks from pixel-level input via autonomous trial-and-error, achieving meaningful in-game progress without game-specific adaptations. These advances underscore the potential of API-free agents to operate in broadly applicable and visually grounded settings, paving the way for more universal and flexible AI systems.

## A.2 MORE METHODOLOGICAL DETAILS

**Prompt Engineering for Agent Reasoning.** Effective reasoning in visual-language models (VLMs) relies on meticulously designed prompts to guide complex visual understanding and decision-making processes Shtedritski et al. (2023); Gu et al. (2023). In KG-Agent, we employ a structured

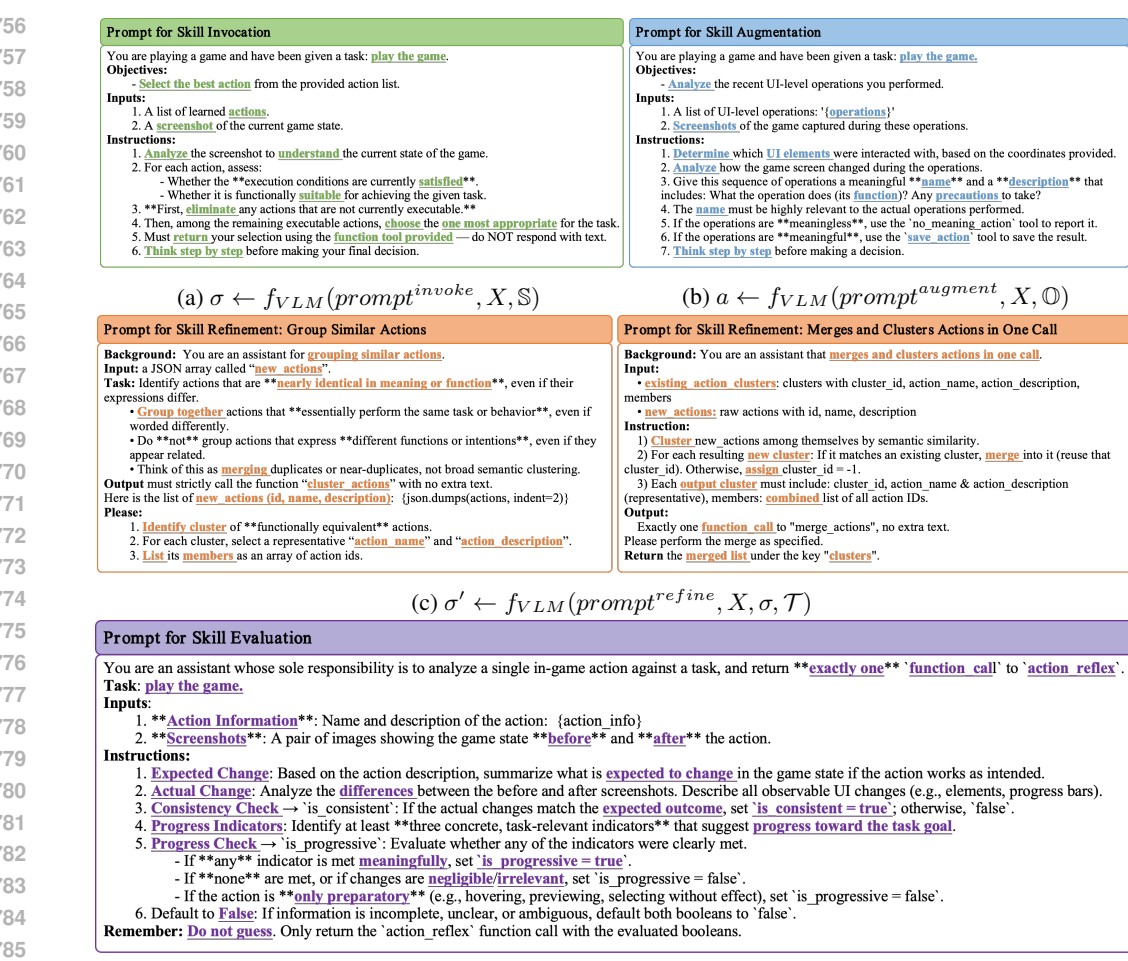

(a) $\sigma \leftarrow f_{VLM}(prompt^{invoke}, X, \mathbb{S})$

(b) $a \leftarrow f_{VLM}(prompt^{augment}, X, \mathbb{O})$

(c) $\sigma' \leftarrow f_{VLM}(prompt^{refine}, X, \sigma, \mathcal{T})$

(d) $\mathcal{R}_{progress}, \mathcal{R}_{semantic} \leftarrow f_{VLM}(prompt^{evaluate}, X_i, X_j, \sigma, \mathcal{T})$

Figure 6: Structured Prompt Design for KG-Agent's VLM Reasoning Stages. (a) Prompt for Skill Invocation enables action selection by analyzing the game state and action conditions. (b) Prompt for Skill Augmentation derives new actions from UI-level operations and screen changes, where $\mathbb{O}$ is the set of operations. (c) Prompt for Skill Refinement (Cluster and Merge) groups to functionally similar new actions. (d) Prompt for Skill Evaluation assesses action effectiveness based on expected versus actual game state changes.

VLM-based reasoning cycle consisting of four key stages: skill invocation, augmentation, refinement, and evaluation. As illustrated in Figure 6, each stage is supported by a dedicated prompt template that collectively enables intelligent interaction in dynamic GUI environments without reliance on pre-defined APIs. Specifically, the $prompt^{invoke}$ in Figure 6a serves as the agent's primary decision-making interface. It selects the most suitable executable action from procedural memory when symbolic reasoning over the SA-kG fails. The $prompt^{augment}$ in Figure 6b complements invocation by deriving new high-level actions from low-level UI operations ($\mathbb{O}$), thereby expanding the agent's skill repertoire based on observed screen changes. The $prompt^{refine}$ in Figure 6c consists of two sub-prompts: one groups functionally similar new actions, and the other merges them with existing skill clusters to maintain a compact and non-redundant skill library. Finally, the $prompt^{evaluate}$ in Figure 6d assesses the effectiveness of executed skills by comparing expected outcomes against actual screen changes, providing essential feedback for progressive and semantic rewards as defined in Eq. 9. Together, these prompts form a closed-loop learning architecture: invocation drives action, augmentation acquires knowledge, refinement consolidates understanding, and evaluation validates progress. This integrated design ensures that KG-Agent can continuously perceive, reason, act, and adapt within complex, unseen GUI environments.

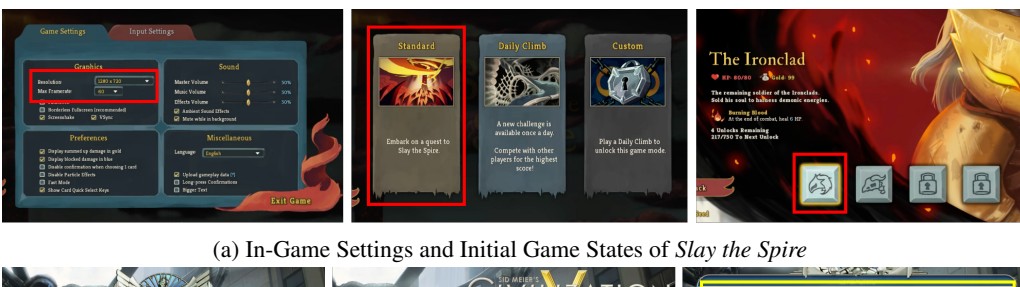

(a) In-Game Settings and Initial Game States of *Slay the Spire*

(b) In-Game Settings and Initial Game States of *Civilization V*

Figure 7: In-game settings and initial game states for *Slay the Spire* and *Civilization V*, maintaining consistent observation conditions and fair evaluation benchmarks.

**Pseudocode of The KG-Agent.** All skill invocation, augmentation, refinement, and evaluation steps follow the algorithmic loop defined in Algorithm 1. We determine node merges through a similarity-based clustering process that operates directly on state feature vectors (Line 4). The implementation follows these steps: 1) Similarity Calculation. For each new state, we compute its feature similarity with all existing nodes using a cosine similarity metric. 2) Threshold-based Decision. If the maximum similarity exceeds our merge threshold ($\theta_{merge} = 0.85$), the state is merged with the most similar node. 3) Feature Update. Merged node features are updated to the mean of the original and new feature vectors, allowing the representation to evolve over time. 4) Graph Maintenance. Similarity edges are automatically updated to maintain the graph's connectivity structure. A critical innovation of our approach is the identification of the *Neighborhood of Experience* $\mathcal{N}_{\mathcal{E}}(v_i)$, which enables the agent to leverage past experiences in similar states. This neighborhood information facilitates the gathering of high-quality candidate skills $\mathbb{S}_{HQ}(v_i)$ from the SA-KG, representing skills that have proven effective in comparable situations. The agent performs multiple attempts to execute them, with each attempt generating reward feedback that updates the skill edges in the SA-KG. The loop breaks early if a sufficiently high reward is achieved, ensuring efficient skill utilization. When no high-quality skills are available, the agent enters a trial-and-reasoning phase. Specifically, in Line 28, the UCT-style selector of MCTS employs a fixed exploration constant C=0.5 and a base exploration utility threshold of 0.1, while dynamically scheduling the temperature parameter $\tau$ through a decay function based on total selection counts of the corresponding skill cluster to gradually shift from exploration to exploitation. Beyond the standard UCT formula, it incorporates several sophisticated threshold mechanisms: 1) a pre-selection filtering mechanism that automatically excludes actions requiring unavailable objects or those in suspended state, 2) an action completeness penalty that proportionally reduces the fitness score based on the ratio of missing prerequisite objects, 3) an exploration dominance trigger that flags pure exploration mode when the calculated probability for exploration actions exceeds 0.9. These specialized thresholds work in concert with numerical stability measures - including max normalization and probability clipping in the softmax calculation - to ensure robust action selection in complex, partially observable environments. This integrated approach enables the KG-Agent to dynamically balance exploitation of proven strategies with exploration of novel solutions, while maintaining an evolving knowledge structure that captures both state relationships and skill effectiveness across the environment.

---

**Algorithm 1** The Pseudocode of Our KG-Agent Skill Evolution

---

**Input**: Environment $(\mathcal{X}, \mathcal{A}, \mathcal{T}, \mathcal{R})$, VLM $f_{VLM}$, Skill library $\mathbb{S} = \emptyset$, and SA-KG $\mathcal{G}(\mathcal{V}, \mathcal{E}) = \emptyset$.

---

1: **for** Each Agent **do**
2:     **while** Episode not terminated **do**
3:         Observe current screen $X_i \in \mathcal{X}$ and extract visual feature $x_i$;
4:         Generate or merge state node $v_i$ in $\mathcal{G}$, connect similarity edges $e^{sim} \in \mathcal{E}^{sim}$;
5:         Identify *Neighborhood of Experience* $\mathcal{N}_{\mathcal{E}}(v_i)$ by Eq. 2;
6:         Gather high-quality candidate skills $\mathbb{S}_{HQ}(v_i)$ from $\mathcal{G}$ by Eq. 3;
7:         **if** $\mathbb{S}_{HQ}(v_i) \neq \emptyset$ **then**
8:             **for** $attempt = 1$ to $max\_attempts$ **do**
9:                 Sample high-quality candidate skill $\sigma_k$ from $\mathbb{S}_{HQ}$ by Eq. 4;
10:                 Observe trajectory $\mathcal{T}$ and compute $\mathcal{R}_{total}(v_i, v_j)$ by Eq. 9;
11:                 Generate skill edges $w_{i,j}^{\sigma_k}$ by Eq. 1;
12:                 **if** $\mathcal{R}_{total}(v_i, v_j) > threshold$ **then**
13:                     **Break**;
14:                 **end if**
15:             **end for**
16:         **else**
17:             Sample candidate skill $\sigma_l$ from $\mathbb{S}_C(v_i)$ by Eq. 7;
18:             **if** $\mathbb{S}_C(v_i) = \emptyset$ **then**
19:                 **for** $k = 1$ to $k_{max}$ **do**
20:                     Generate $a \leftarrow f_{VLM}(prompt^{augment}, X_i, \mathbb{O})$;
21:                     **if** $a$ is $recognizable$ **then**
22:                         Update Skill library $\mathbb{S}$;
23:                         **break**;
24:                     **end if**
25:                 **end for**
26:             **else**
27:                 Select the best skill cluster by $f_{VLM}(prompt^{invoke}, X, \mathbb{S})$;
28:                 Evaluate the skill cluster with MCTS;
29:                 Execute the best skill $\sigma$ from MCTS;
30:             **end if**
31:             Observe trajectory $\mathcal{T}$ and compute $\mathcal{R}_{total}(v_i, v_j)$ by Eq. 9;
32:             **if** $\mathcal{R}_{total}(v_i, v_j) < threshold$ **then**
33:                 Remove $\sigma$ from $\mathbb{S}$;
34:             **end if**
35:             **if** $skill\_count < threshold$ **then**
36:                 Refine skill: $\sigma' \leftarrow f_{VLM}(prompt^{refine}, X, \sigma, \mathcal{T})$;
37:                 Replace $\sigma$ with $\sigma'$ if improvement observed;
38:             **end if**
39:         **end if**
40:     **end while**
41: **end for**

---

## A.3   More Experimental Setting

**Open-ended Environment.** As shown in Figure 7, we evaluate the KG-Agent in two challenging games *Slay the Spire* (base difficulty: Ascension 0, character: Ironclad), which blends roguelike and deck-building mechanics, and *Civilization V* (Russia, Earth map, Standard size, Prince difficulty, Standard pace), a hallmark 4X (eXplore, eXpand, eXploit, eXterminate) strategy game. The failure/termination policy is governed by two primary conditions. First, an upper bound of 500 steps is enforced for any single episode to maintain computational efficiency. Second, and more critically, early termination is triggered by task-specific failure conditions. In *Slay the Spire*, this occurs when the character being defeated by the monster. In *Civilization V*, termination is triggered upon the nation being defeated by the AI. This dual-layer policy ensures that training efficiently halts upon

Table 4: Sensitivity analysis of key hyperparameters in *Slay the Spire*, including the number of skills sampled per execution attempt $M$, the coefficient balancing state-value reward $\mathcal{R}_{state}$ and novelty reward $\mathcal{R}_{novel}$, and the cosine-similarity thresholds for node merging $\theta_{merge}$ and similarity edge creation $\theta_{simi}$. Nodes Size, Skill Edges, and Similarity Edges denote the number of nodes, skill edges, and similarity edges in the SA-KG, respectively. Floors cleared and official run scores indicate the in-game advancement. Runtime Per Step is the average time in seconds required to run one simulation step under each configuration.

| Hyperparameters | Configuration | Nodes Size | Skill Edges | Simi. Edges | Floors ↑ Cleaned | Run ↑ Scores | Runtime↓ Per Step |
|---|---|---|---|---|---|---|---|
| Number of Skills Sampled Per Attempt | $M = 5$ | 29 | 54 | 354 | **16** | **112** | **158.34s** |
| | $M = 3$ | 25 | 38 | 124 | **16** | 98 | 244.63s |
| | $M = 10$ | 22 | 34 | 116 | 11 | 63 | 192.41s |
| Reward Coefficient ($\mathcal{R}_{state} : \mathcal{R}_{novel}$) | 1 : 1 | 29 | 54 | 354 | **16** | **112** | 158.34s |
| | 1 : 2 | 33 | 32 | 220 | **16** | 102 | 153.26s |
| | 2 : 1 | 20 | 26 | 114 | 13 | 87 | **145.92s** |
| Cosine Threshold ($\theta_{merge}, \theta_{simi}$) | 0.95, 0.88 | 29 | 54 | 354 | **16** | **112** | **158.34s** |
| | 0.99, 0.90 | 155 | 190 | 14520 | 12 | 70 | 392.55s |
| | 0.85, 0.80 | 4 | 9 | 4 | 4 | 26 | 179.40s |

either reaching the step limit or encountering a definitive in-game failure state. Both games reflect the core challenge of our problem setting: they lack high-level programmatic interfaces and require agents to interact exclusively through pixel-based GUIs. This stands in sharp contrast to embodied agent frameworks such as VOYAGER Wang et al. (2023a) and SkillWeaver Zheng et al. (2025), which leverage structured APIs (e.g., Mineflayer) for motor control, enabling a focus on high-level, lifelong learning while abstracting away low-level perception and control. In our API-free setting, agents cannot rely on predefined action primitives or semantic state abstractions typically provided by an API. Instead, they must reason directly over raw visual observations, introducing significant challenges for *exploration efficiency* and *long-horizon planning*. To assess whether existing skill-library-based methods could adapt to this setting, we reproduced the prompt engineering and skill retrieval strategies from VOYAGER within our baseline framework. However, both approaches failed to achieve meaningful in-game progress, underscoring their heavy reliance on structured APIs and the limited transferability of their learned skills to purely visual interaction scenarios. While KG-Agent shares the same high-level paradigm, i.e., constructing executable skills on the fly and retrieving relevant ones for execution, with VOYAGER, the absence of an API is not a superficial difference in interaction modality. Rather, it constitutes a fundamental shift in the learning problem, requiring new mechanisms capable of overcoming the limitations of pixel-based environments. To this end, our proposed SA-KG is designed to address key challenges such as myopic decision-making and inefficient trial-and-error learning, enabling more effective reasoning and skill reuse in API-free GUI-based settings.

## A.4 MORE EXPERIMENTAL RESULTS

**Hyperparameter Sensitivity.** While tuning hyperparameters is often unavoidable in complex multi-agent systems Du et al. (2025); Tan et al. (2025); Wang et al. (2023a); Zheng et al. (2025), we emphasize that our KG-Agent employs the same set of hyperparameters, prompts, and experimental setups across different game environments, underscoring the generalizability of our method and reducing the need for extensive manual tuning when adapting to new domains. For certain hyperparameters, e.g., the criterion for increasing skill length, the fitness sensitivity and greediness weighting constants, and the temperature for skill selection, we adopt values grounded in established practices from related work Du et al. (2025). Furthermore, for the key hyperparameters directly tied to our core contributions, i.e., the coefficient balancing state-value and novelty rewards, the cosine-similarity thresholds for node merging and edge creation, and the number of skills sampled per execution

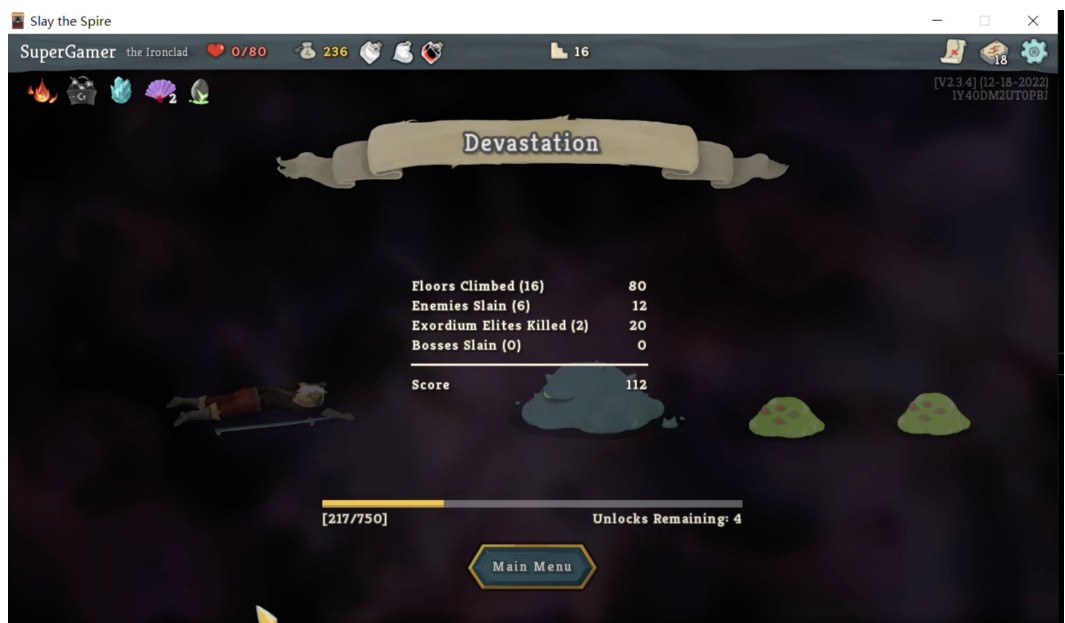

Figure 8: In-game advancement measured by floors cleared and official run score in *Slay the Spire*.

attempt, we conduct a targeted sensitivity analysis in *Slay the Spire*. As summarized in Table 4, for each of these key parameters, we evaluated two alternative values alongside the default setting, while keeping all other configurations identical.

As shown in Table 4, our selected hyperparameter configuration—sampling a maximum of 5 skills per execution attempt ($M = 5$), applying a balanced reward coefficient ($\mathcal{R}_{state} : \mathcal{R}_{novel} = 1 : 1$), and setting cosine similarity thresholds for node merging and edge creation to $\theta_{merge} = 0.95$ and $\theta_{simi} = 0.88$, stands out as the optimal setup. It achieves superior in-game performance, attaining the highest score of 112 and maximum progression of 16 floors cleared, while preserving high computational efficiency with an average runtime of 158.34 seconds per step. The choice of $M = 5$ strikes an ideal trade-off between behavioral diversity and execution efficiency, outperforming both smaller and larger sampling sizes. Similarly, the 1:1 reward ratio proves more effective than imbalanced configurations, better harmonizing exploration and exploitation compared to either novelty-heavy ($\mathcal{R}_{state} : \mathcal{R}_{novel} = 1 : 2$) or state-value-dominated ($\mathcal{R}_{state} : \mathcal{R}_{novel} = 2 : 1$) settings. Importantly, the model exhibits robust performance across a range of configurations, with multiple hyperparameter sets yielding solid results, underscoring its general insensitivity to parameter variation, though our specific configuration delivers the best balance between task advancement and operational efficiency.

It's worth noting that the alternative cosine thresholds performed poorly (as shown in the last two rows of Table 4). Specifically, excessively strict thresholds ($\theta_{merge} = 0.99$, $\theta_{simi} = 0.90$) result in an overly dense knowledge graph, with nodes and similarity edges proliferating to 155 and 14,520, respectively. This structural complexity severely degrades runtime performance (392.55 seconds/step) and fragments the state space, thereby hindering skill generalization and limiting progression to only 12 floors. On the other hand, overly relaxed thresholds ($\theta_{merge} = 0.85$, $\theta_{simi} = 0.80$) yield an extremely sparse graph with merely 4 nodes and 4 similarity edges, effectively disabling the SA-KG mechanism by preventing meaningful skill associations. This oversimplification also compromises the reliability of novelty rewards and state-value estimates, misleading the agent's decision-making and restricting progress to just 4 floors. In contrast, our chosen thresholds ($\theta_{merge} = 0.95$, $\theta_{simi} = 0.88$) strike an optimal balance: 354 similarity edges ensure sufficient connectivity to support skill transfer, while 29 nodes maintain a tractable graph scale. This balanced structure underpins the agent's top performance, i.e., 16 floors cleared and an in-game score of 112.

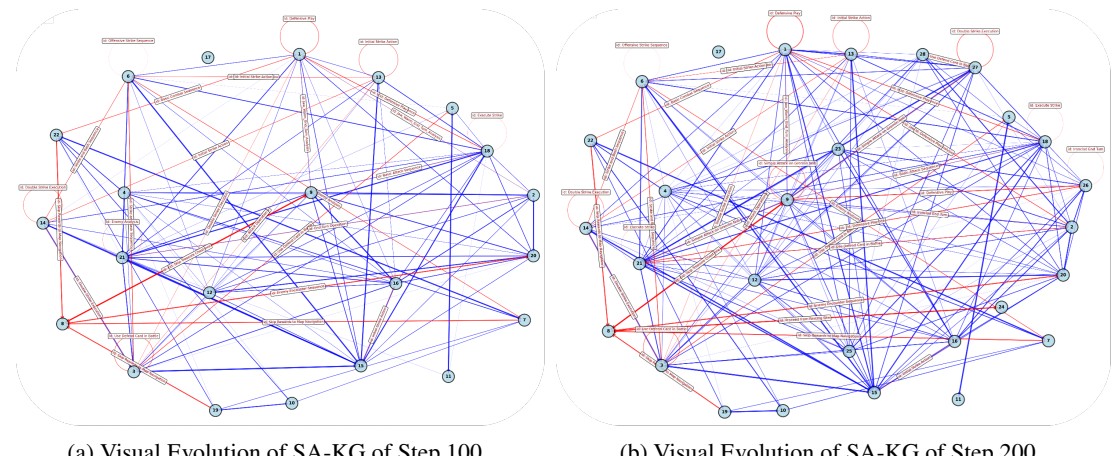

| (a) Visual Evolution of SA-KG of Step 100 | (b) Visual Evolution of SA-KG of Step 200 |

Figure 9: The State-Action Knowledge-Graph (SA-KG) evolution from step 100 to 200 shows network growth (22 → 28 nodes, 190 → 312 similarity edges, and 37 → 51 skill edges), with skill (red) and similarity (blue) connections reflecting improved structural organization and relational reasoning capabilities.

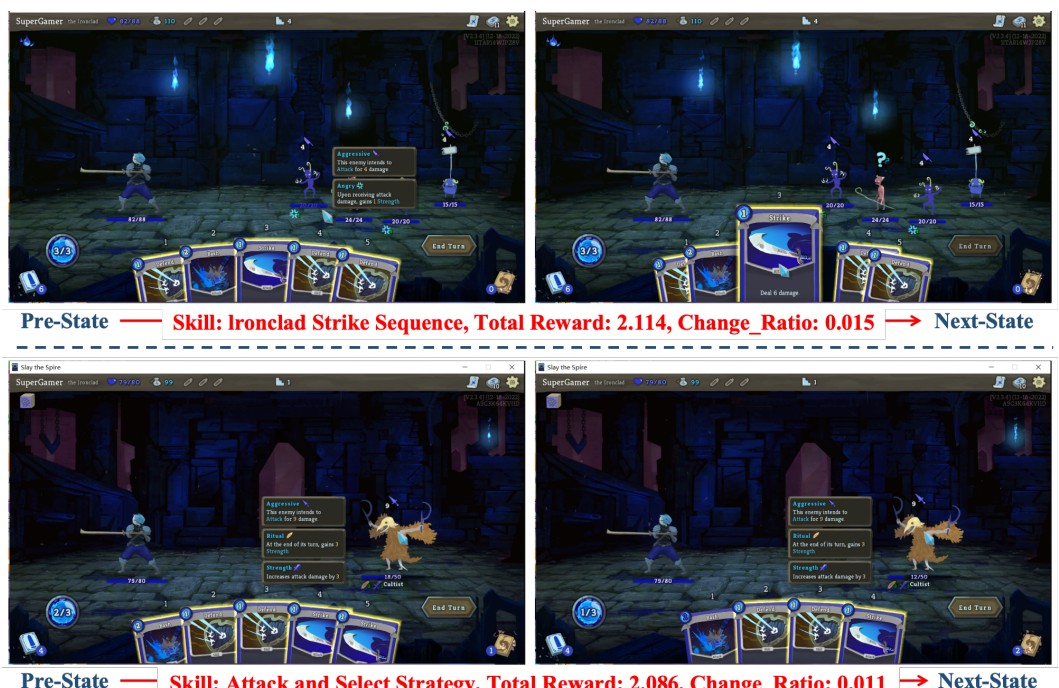

Figure 10: Skill invocation with low visual change but high reward in *Slay the Spire*.

**Generalization to Slay the Spire: A Zero-Shot Transfer.** The following experiment serves as a stringent test of our framework's generalization capability through its zero-shot transfer to *Slay the Spire*. This evaluation deploys an identical instance of the system, utilizing the same architecture, prompts, and hyperparameters as those used in *Civilization V*. No component of the system was modified, fine-tuned, or domain-specifically adjusted for this new domain. Consequently, the results provide direct evidence of its inherent adaptability and robustness. Figure 8 illustrates the in-game advancement measured by floors cleared and official run score in *Slay the Spire*. Figure 9 illustrates the structural evolution of the SA-KG from step 100 to step 200, with the evaluation terminating at step 210 after the agent's character is defeated by a monster. Over this interval, the knowledge graph

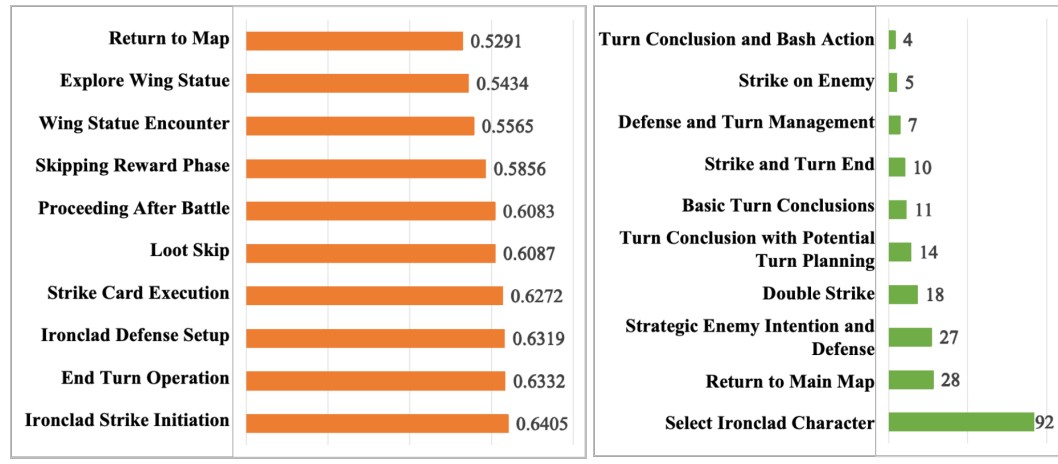

(a) Skill Edges: Top 10 by Weight          (b) Skill Cluster: Top 10 by Number

Figure 11: The structured skill acquisition of KG-Agent in *Slay the Spire*, demonstrating how the SA-KG framework overcomes pixel-level myopia. (a) High-utility skill transitions showing optimized combat sequences learned through the hybrid reward mechanism. (b) Frequently invoked skill clusters revealing hierarchical organization from basic operations to strategic planning, evidencing effective generalization across visually distinct GUI states.

undergoes substantial growth: the number of nodes increases from 22 (step 100) to 28 (step 200), similarity edges rise from 190 (step 100) to 312 (step 200), and skill edges grow from 37 to 51. This expansion reflects advancing relational reasoning capabilities—particularly the accelerated formation of similarity edges, which signal improved recognition of state equivalences and environmental patterns. Meanwhile, the steady increase in skill edges suggests effective knowledge transfer and progressive maturation of the graph, driven by denser connectivity and the emergence of cohesive clusters that support more robust decision-making.

Figure 10 demonstrates our agent's capacity in *Slay the Spire* to prioritize strategic value over perceptual salience, as evidenced by the selection of combat skills like "Ironclad Strike Sequence" (Total Reward: 2.114, Change_Ratio: 0.015) and "Attack and Select Strategy" (Total Reward: 2.086, Change_Ratio: 0.011) that yield consistent rewards despite minimal visual changes between states. This reflects the agent's advanced reasoning capabilities in identifying strategically valuable actions—such as optimal card sequencing and targeted enemy selection—that contribute to incremental combat advantage and resource management, even when immediate visual feedback is subtle. The ability to transcend superficial interface cues and consistently execute high-value tactical decisions underscores the framework's sophistication in modeling complex game dynamics, where subtle but strategically sound actions accumulate toward long-term success in challenging roguelike environments. In addition, Figure 11 also demonstrates KG-Agent's effectiveness in overcoming the key limitations of API-free GUI agents by structuring pixel-level interactions into the SA-KG. The high-utility skill edges, e.g., combat sequences and post-battle navigation, show the agent's ability to prioritize strategic actions beyond immediate rewards, overcoming myopic decision-making. Simultaneously, the frequent invocation of abstract skill clusters, particularly the dominant "Select Ironclad Character" with 92 invocations, demonstrates successful generalization across visually distinct states, enabling efficient long-term planning by recalling proven strategies rather than relying on repetitive pixel-level exploration.

## A.5 CASE STUDY

Here we present two representative case studies from the game *Slay the Spire* to illustrate further how KG-Agent structures pixel-based interactions into reusable skills and enables strategic generalization through its SA-KG.

Figure 12 presents a case study demonstrating successful skill reuse through the SA-KG of KG-Agent. When encountering a new state and corresponding node $v_3$, the proposed KG-Agent

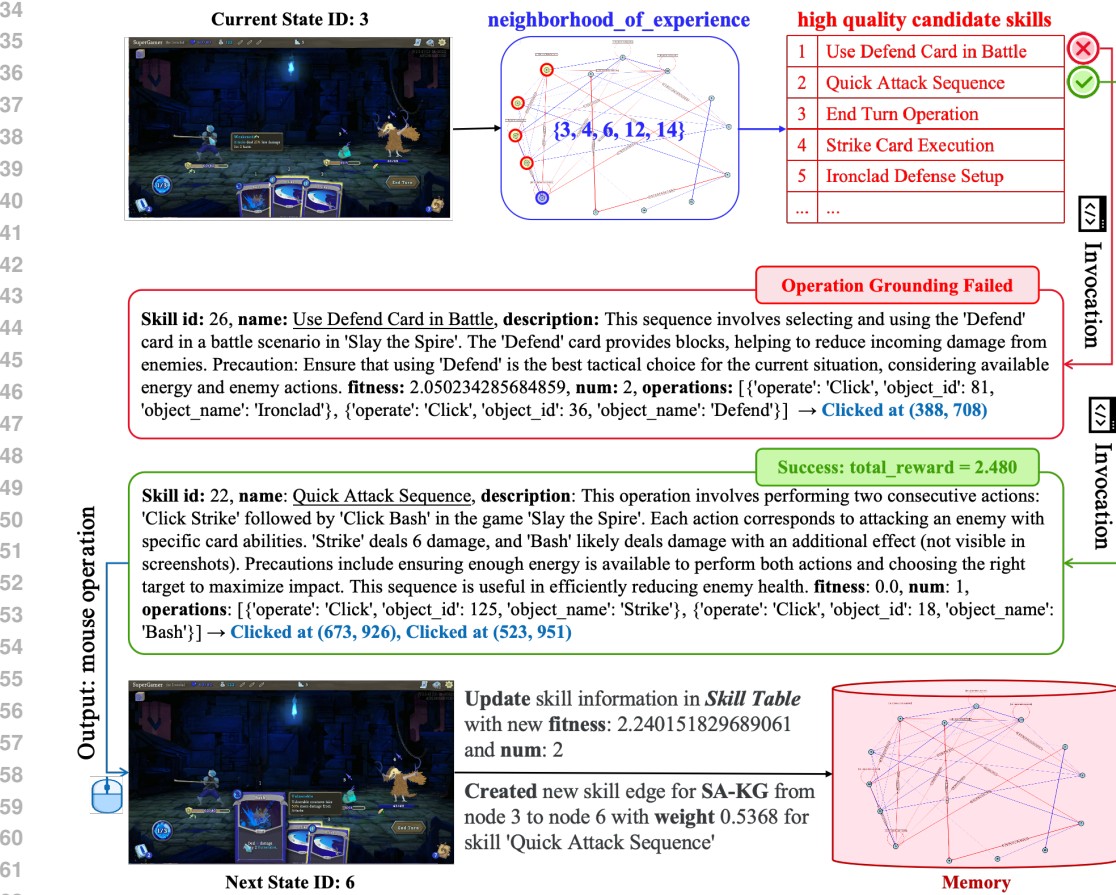

Figure 12: Case study of successful skill invocation through state-conditioned sampling from the SA-KG. Given the current state, KG-Agent retrieves high-quality candidate skills from the neighborhood of experience $\mathcal{N}_{\mathcal{E}}$ and samples skills according to the weights of skill edges. While the first sampled skill *"Use Defend Card in Battle"* fails due to grounding issues, the second skill *"Quick Attack Sequence"* executes successfully. The agent subsequently updates the skill's fitness and reinforces its edge weight in the Memory.

first retrieves its *Neighborhood of Experience* $\mathcal{N}\mathcal{E}(v_3)$ and identifies high-quality candidate skills $\mathbb{S}HQ(v_3)$. Skills are then sampled probabilistically according to their skill edge weights in the SA-KG. While the initially sampled skill "*Use Defend Card in Battle*" fails due to grounding issues, the subsequently sampled skill "*Quick Attack Sequence*" executes successfully through the simulated mouse operations "*Clicked at (673, 926)*" and "*Clicked at (523, 951)*", achieving a total reward of 2.480 and transitioning to next state node $v_6$. Following successful execution, KG-Agent updates both the fitness value and edge weight of this skill in Memory, thereby reinforcing this validated strategy for future reuse. The case study in Figure 12 illustrates how KG-Agent addresses the key limitations of traditional LLM-based agents in API-free environments. First, by organizing pixel-level interactions into SA-KG, the agent overcomes inefficient exploration by generalizing across visually distinct but functionally similar states through *Neighborhood of Experience*, avoiding myopic decisions. Second, the edge-weight based sampling demonstrates strategic reuse of historical knowledge rather than relying on trial-and-error. Third, the successful execution and subsequent memory update illustrate our hybrid reward mechanism: the substantial environmental reward combined with fitness updates reinforces high-value pathways while maintaining exploration flexibility. This showcases KG-Agent's ability to decouple strategic planning from pure discovery, effectively addressing both skill acquisition and long-horizon reasoning challenges in API-free environments.

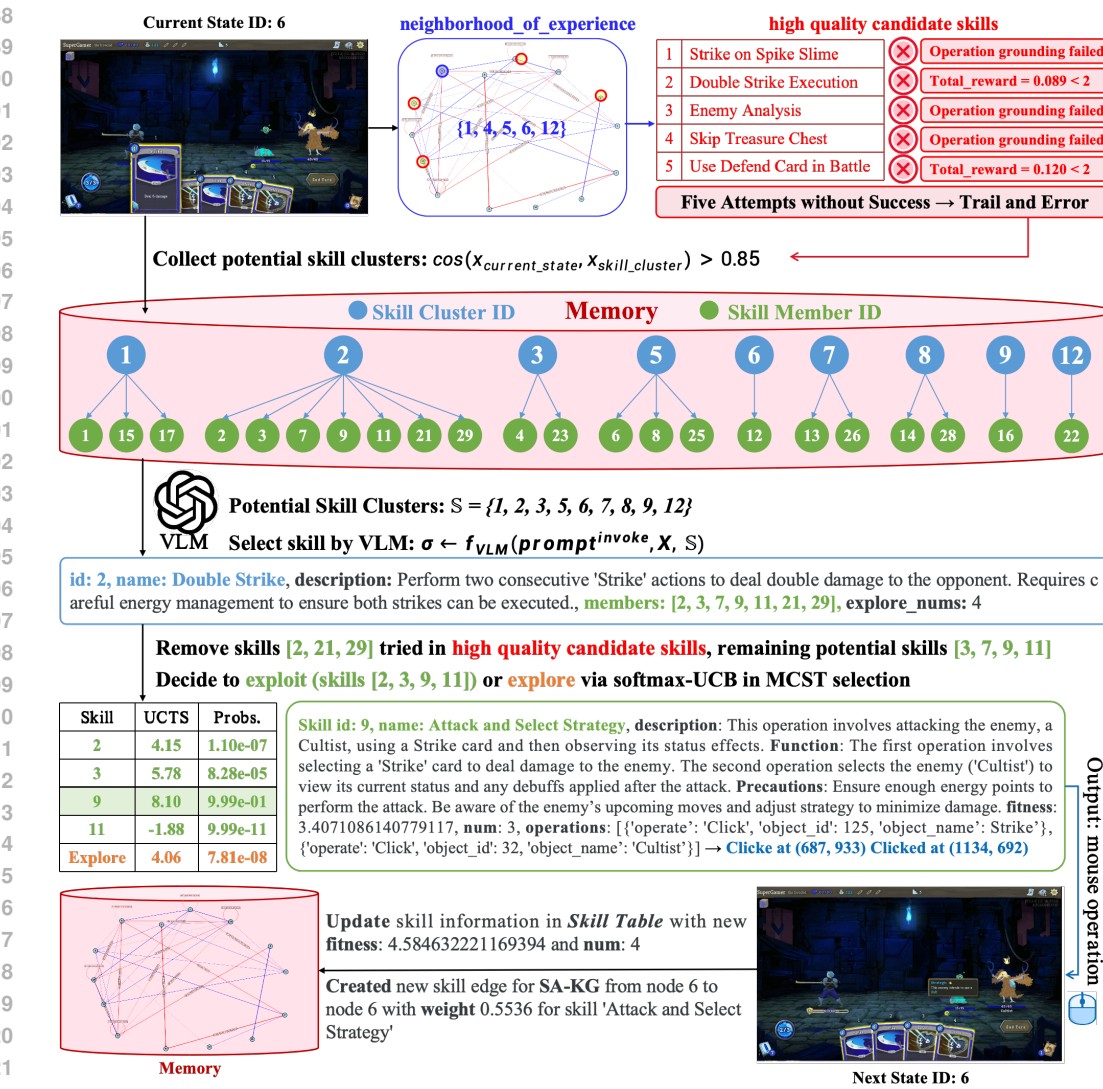

Figure 13: Case study of successful skill invocation by general VLM-guided trial and error. After five unsuccessful attempts with SA-KG-sampled skills, KG-Agent switches to trial-and-error mode. It first collects skill clusters from Procedural Memory using VLM-based selection, then applies MCTS with UCTS strategy to balance exploration and exploitation. The successfully executed skill "*Attack and Select Strategy*" is subsequently updated in memory with revised fitness and edge weight.

Figure 13 demonstrates KG-Agent's two-stage skill invocation strategy, which prioritizes structured knowledge from the SA-KG before resorting to VLM-guided search. Initially facing state node $v_6$, the KG-Agent attempts SA-KG-based skill sampling but encounters five consecutive failures due to grounding issues or insufficient rewards. This triggers a fallback to VLM-guided skill retrieval from *Procedural Memory*. The process first queries the *Action Clusters Table* to obtain candidate skill clusters $\mathbb{S}_C(v_6) = \{1, 2, 3, 5, 6, 7, 8, 9, 12\}$, from which the VLM identifies the most contextually relevant cluster based on visual observation. To navigate the stochastic environment, the KG-Agent employs a UCT-inspired strategy that balances exploitation of high-value skills $\{\sigma_2, \sigma_3, \sigma_9, \sigma_{11}\}$ with exploration of less-tried options. The resulting utility scores are converted into a probability distribution via temperature-scaled softmax, enabling stochastic sampling. The first sampled skill "*Attack and Select Strategy*" ($Probs. = 9.99e - 01$) executes successfully through operations "*Clicked*

*at (687, 933)*" and "*Clicked at (1134, 692)*", yielding a 2.117 reward. This successful execution triggers updates to the skill's fitness and edge weight in memory, demonstrating KG-Agent's capacity for continuous knowledge refinement through both successful and failed experiences.

## A.6 LARGE LANGUAGE MODELS USAGE STATEMENT

In accordance with the conference policy on the use of LLMs, we disclose that an LLM was used solely as a general-purpose assistive tool for grammatical refinement and language polishing of the manuscript. The LLM played no role in research ideation, experimental design, data analysis, interpretation of results, or substantive writing. All scientific content, claims, and technical contributions were conceived, developed, and verified exclusively by the human authors. The authors take full responsibility for the accuracy, integrity, and originality of the work.

