# OpenReview forum: "Experience-Driven Exploration for Efficient API-Free AI Agents"
_ICLR.cc/2026/Conference — Submitted to ICLR 2026_

### Official Review · Reviewer_qQ3q · 2025-10-23

**Soundness:** 3
**Presentation:** 2
**Contribution:** 3
**Rating:** 6
**Confidence:** 3

**Summary:**

The paper focuses on gaming agents, and proposes a framework based on GUI-agents to improve their performance on games such as Civilization 5 and Slay the Spire. Specifically, the authors propose KG-agent, which utilizes a knowledge-graph like skill library to tackle long horizon games. On a high level, the proposed approach uses existing/related skills from the knowledge graph when given a new game state (represented as a screenshot) if similar states can be found, or uses the VLM model directly as GUI agents to generate new actions and add new skills to the graph. Experimental results on two games shows the effectiveness of the proposed method compared to its prior work (BottomUp).

**Strengths:**

- Introducing a graph structure for skill libraries is intuitive and novel.

- Experiments on two different games (Civilization V and Slay the Spire) shows the effectiveness of the proposed method, improving both game score and token cost compared to its prior work (BottomUp).

- The authors conducted numerous analysis of the method, including visualizing the evolution of the skills graph and an ablation study on the proposed method.

**Weaknesses:**

1. The notion of using a skill library for GUI related tasks highly resembles existing work such as [1] and [2]. However, there is no comparison against these simpler skill-library based methods.

2. The proposed method includes a lot of heuristics and modules (four tables for procedure memory, a state-action KG, a reasoning module, along with 11 equations). It was challenging to understand the real contribution of each of these items, especially when the ablation study in Table 1 only ablated two aspects: rewards (covering equation 9-11), and similarity edges in the KG (covering equation 2-5). A lot of other design choices such as having four tables in the procedure memory, important of the reasoning module, as well as the effect of various hyperparameters/equation choices were omitted in the paper.


---

References

[1] Wang, Guanzhi, et al. "Voyager: An open-ended embodied agent with large language models." arXiv preprint arXiv:2305.16291 (2023).

[2] Zheng, Boyuan, et al. "Skillweaver: Web agents can self-improve by discovering and honing skills." arXiv preprint arXiv:2504.07079 (2025).

**Questions:**

- Section 3.1 introduces a method using SAM to help VLMs to generate executable actions on an GUI image. I wonder if the authors have tried/are aware of VLMs such as OpenAI's operator and Claude's computer-use, which can directly output low-level actions on an GUI image without using additional models such as SAM.

- L424 and Table 3 shows that removing similarity edges from the graph impairs long-term progression/planning. What is the intuitive reason behind this? To me, it seems to indicate that the backbone VLMs themselves are bad at decision-making, and programmatic approach such as reusing skills from similar states is more robust than querying the VLMs?

- How are the hyperparameters such as theta, alpha, $C_0$, etc selected on L376-377? These values seems very specific.

---

> ### Author Response · Authors · 2025-11-21
> **Response (1/2)**
>
> We are truly grateful for the time you have taken to review our paper, your insightful comments, and support. Your positive feedback is incredibly encouraging for us! In the following response, we would like to address your major concern and provide additional clarification.
>
> **W1. Baseline Comparison**
>
> **A1:** We thank the reviewer for this comment. Our KG-Agent is fundamentally different from these simpler skill-library-based methods, e.g., Voyager and SkillWeaver, which largely rely on high-level APIs for skill generation and invocation. In contrast, our method is specifically designed to operate in **API-free setting**. As explicitly stated in Voyager:
> >... we do not directly compare with methods that take screen pixels as input and output low-level controls. It would **not be an apple-to-apple comparison**, because we rely on API to...
>
> Similarly, comparing our method with methods rely on APIs does not constitute a fair comparison.
>
> **W2 & Q3. Implementation Details**
>
> **A2:** Thank you for this critical feedback. The four tables form an integrated cognitive architecture, not merely disable one component. While theoretically consolidatable, a single table would significantly increase retrieval complexity and cost. Our modular design intentionally separates knowledge types to enable efficient, contextual querying of small relevant subsets.
>
> We conduct a new ablation study on VLM-based Rewards ($w/o$ $R_{progress}$ & $R_{semantic}$) in Table 3. The severe performance drop confirms that the absence of these high-level reward signals, which are generated by our reasoning module, leads to a catastrophic failure in task performance, underscoring the reasoning module's critical role.
>
> | Setting | Library Size | Node Size | Skill Edge | Simi. Edge | Turns Survived | Techs. Unlock | Response. Rate | Token Costs |
> |------------------------------------|--------------|-----------|------------|------------|-----------------|---------------|---------------------------|--------------|
> | Full KG-Agent Model | 76 | 26 | 37 | 226| 65 | 10 | 0.89 | 3.0 |
> | w/o $R_{novel}$ & $R_{state}$ | 64 | 18 | 39 | 96 | 35 | 3 | 0.79 | 3.3 |
> | w/o $R_{progress}$ & $R_{semantic}$ | 33 | 9| 14 | 42 | 14 | 1 | 0.59 | 2.1 |
>
> Moreover, for the key hyperparameters directly tied to our core innovations, i.e.，the number of skills sampled ($M$), the reward coefficient ratio (RC = $R_{state}$ : $R_{novel}$), and the merge/similarity thresholds ($\theta_{merge}$, $\theta_{simi}$), we conducted a targeted sensitivity analysis on Slay the Spire in Table 4 (see Appendix).
>
> | Config. | Node Size | Skill Edge | Simi. Edge | Floor Clear | Run Score | Runtime|
> --------------|------------|-------------|-------------|---------------------|-----------------|------------------|
> | *M* = 5 | 29 | 54 | 354 | 16 | 112 | 158.34 |
> | *M* = 3 | 25 | 38 | 124 | 16 | 98 | 244.63 |
> | *M* = 10 | 22 | 34 | 116 | 11 | 63 | 192.41 |
> | RC = 1 : 1 | 29 | 54 | 354 | 16 | 112 | 158.34 |
> | RC = 1 : 2 | 33 | 32 | 220 | 16 | 102 | 153.26 |
> | RC = 2 : 1 | 20 | 26 | 114 | 13 | 87 | 145.92 |
> | 0.95, 0.88 | 29 | 54 | 354 | 16 | 112 | 158.34 |
> | 0.99, 0.90 | 155 | 190 | 14520 | 12 | 70 | 392.55 |
> | 0.85, 0.80 | 4 | 9 | 4 | 4 | 26 | 179.40 |
>
> The model shows robust performance across configurations, with our chosen setup providing the optimal balance between task progress and operational efficiency.
>
> We are truly grateful for this suggestion, which significantly strengthens our work. In our revised version, we have significantly expanded our experiments to address these points directly.

---

> ### Author Response · Authors · 2025-11-21
> **Response (2/2)**
>
> **Q1. Necessity of SAM**
>
> **A3:** Thank you for this important observation. We are aware of end-to-end VLMs like Operator and Computer-Use, which represent a promising future direction. However, our modular architecture prioritizes computational efficiency and operational stability for long-horizon tasks—a design philosophy shared by other leading API-free GUI agents (e.g., Bottom-up-Agent, Cradle). Our approach uses CLIP to provide stable state embeddings, ensuring graph consistency over hundreds of steps, unlike dynamically-conditioned planner VLMs whose varying representations would disrupt long-term coherence. For frequent visual validations like "visual change" detection, SAM with template matching offers superior precision and lower computational cost versus repeated VLM queries, making it essential for sustainable knowledge graph construction. We have added discussion of these trade-offs in the manuscript, positioning our method as a practical solution for current environments while recognizing end-to-end models as valuable future work.
>
> **Q2. Results Analysis**
>
> **A4:** Thank you for this insightful question. The performance drop in the '$w/o$ similarity edge' ablation stems from a critical architectural coupling: while we removed state merging, we retained the graph-based rewards $R_{novel}$ and $R_{state}$. This created a pathological incentive: with merging disabled, every state became a new node, so $R_{novel}$ rewarded graph expansion rather than semantic discovery. Simultaneously, $R_{state}$ operated on a fragmented graph, producing noisy value estimates. The VLM's reasoning was thus misled by corrupted rewards, not by inherent failure. This result demonstrates that similarity edges are essential infrastructure—enabling state abstraction and stable value estimation—and that our components form a synergistic architecture, not isolated modules. We have added this analysis to the new version.
>
> Thanks again for appreciating our work and for your constructive suggestions. We have properly included all the rebuttal contents in the revised version, following your valuable suggestions.

---

> ### Comment · Reviewer_qQ3q · 2025-11-25
>
> Thanks for the response.
>
> > Response to W1: ... Similarly, comparing our method with methods rely on APIs does not constitute a fair comparison.
>
> I am uncertain about this. My point in my original concern was that all these methods are based on a skill library generated on-the-fly, similar to yours. To construct such a skill library, works such as Voyager and SkillWeaver depends on APIs as the "low-level" actions, whereas this work depends on GUI actions as the "low-level" actions. *However, I believe this difference is minor,* as both are essentially callable/executable functions in the environment (e.g., the candidate skills in your figure 2). Therefore, I believe comparisons are still valid since these prior work represent simpler paradigms of 1) constructing these executable functions on-the-fly; 2) locating/retrieving relevant functions to execute.
>
>
> > Response to W2
>
> Thanks. I believe the added results resolved my concern.

---

> > ### Author Response · Authors · 2025-11-26
> > **Response to Comment on Experimental Comparison**
> >
> > Thank you for this valuable insight. We fully agree that a comparison with skill-library-based methods like Voyager and SkillWeaver is valid. We therefore commit to adding this analysis to our experimental section in the final version. As reproducing these baselines will take a few extra days, we will strive to incorporate the comparative results by December 3 deadline.

---

### Official Review · Reviewer_V6a9 · 2025-10-28

**Soundness:** 2
**Presentation:** 3
**Contribution:** 2
**Rating:** 4
**Confidence:** 4

**Summary:**

The paper proposes KG-Agent, an API-free GUI agent that structures pixel-level interaction history into a State–Action Knowledge Graph (SA-KG). Nodes are CLIP-encoded GUI states; edges are either (i) similarity edges linking functionally similar but visually different states or (ii) skill edges recording successful multi-step skills. A hybrid intrinsic reward combines (a) a potential-based state-value term computed from SA-KG out-edge weights and (b) a novelty bonus for first-seen states. The agent uses SAM for interactable-object discovery, template matching for grounding, and a VLM-driven loop for skill invocation/augmentation/refinement with a UCT-style selector. Experiments on Slay the Spire and Civilization V report higher in-game progression and lower token cost than baselines, plus ablations removing similarity edges and reward terms.

**Strengths:**

1.The paper targets an important and difficult setting where agents must learn from pixels with mouse/keyboard only.

2.Converting episodic pixel experience into a connected SA-KG is conceptually clean and practically useful for retrieval and reuse; the “neighborhood of experience” nicely mitigates myopic nearest-neighbor retrieval.

**Weaknesses:**

1.The paper argues that turning experience into a state–action knowledge graph (SA-KG) boosts exploration and skill reuse, but it doesn’t really explain why this should beat simpler memories (like episodic lookup or nearest-neighbor with a bit of temporal logic) or a lightweight model-based rollout. Some clean, controlled head-to-head tests would make that claim much more convincing.

2.The high-level story of how the SA-KG, edges, and hybrid reward work is clear enough, but key nuts-and-bolts are missing: what exactly is the “visual change” metric, how are node merges decided, how is edge “fitness” updated over time, what’s the failure/termination policy, and what thresholds or schedules does the UCT-style selector actually use?

3.The current ablations (dropping similarity edges or intrinsic-reward terms) don’t really hit the core of the method. What’s needed is module-wise ablation: swap the graph memory for strong non-graph baselines under the same perception/grounding, vary graph topology and merge/similarity thresholds, and check sensitivity to graph size/pruning and encoder choice.

4.The system may be more complicated than it needs to be. Using a separate, frozen CLIP for state similarity instead of the planner’s own VLM embeddings risks a representational mismatch and extra moving parts. On the action side, leaning on SAM and template matching for interactable discovery runs counter to the recent trend toward direct VLM grounding—i suggest the author should show it’s actually safer, cheaper, or more reliable at comparable quality.

**Questions:**

See Weaknesses Part

---

> ### Author Response · Authors · 2025-11-21
> **Response (1/2)**
>
> We are truly thankful for your insightful and constructive review. Our detailed responses are presented below.
>
> **W1. Memory Structure**
>
> **A1:** Thank you for this insightful comment. Our experiments already highlight a key distinction: while the Bottom-up Agent relies on episodic lookup and skill replay, the KG-Agent's SA-KG enables skill composition and state abstraction for genuine generalization. As Table 1 shows, KG-Agent significantly outperforms the baseline, which plateaus due to its inability to adapt skills to novel situations. The performance collapse in the "w/o similarity edge" ablation (Table 3) further confirms that graph connectivity—not mere experience storage—is crucial for sustained progress.
>
> **W3. Ablation Studies**
>
> **A3:** Thank you for this insightful comment. We directly address this through our comparison with the Bottom-up Agent—a strong non-graph baseline using episodic skill replay without state abstraction. The substantial performance gap (Table 1) demonstrates the fundamental superiority of our SA-KG for long-horizon tasks, enabled by its state abstraction and relational reasoning. Additionally, our new ablation study (Table 3, $w/o$ $R_{progress}$ & $R_{semantic}$) confirms the graph's dependence on high-level semantic grounding, as their removal causes severe performance degradation.
>
> | Setting | Library Size | Node Size | Skill Edge | Simi. Edge | Turns Survived | Techs. Unlock | Response. Rate | Token Costs |
> |------------------------------------|--------------|-----------|------------|------------|-----------------|---------------|---------------------------|--------------|
> | Full KG-Agent Model | 76 | 26 | 37 | 226| 65 | 10 | 0.89 | 3.0 |
> | w/o $R_{novel}$ & $R_{state}$ | 64 | 18 | 39 | 96         | 35 | 3 | 0.79 | 3.3  |
> | w/o $R_{progress}$ & $R_{semantic}$ | 33 | 9| 14 | 42 | 14 | 1 | 0.59 | 2.1 |
>
> In addition, for the key hyperparameters directly tied to our core innovations, i.e.，the number of skills sampled (M), the reward coefficient ratio (RC = $R_{state}$ : $R_{novel}$), and the merge/similarity thresholds ($\theta_{merge}$, $\theta_{simi}$), we conducted a targeted sensitivity analysis on *Slay the Spire* in Table 4, which is included in the Appendix of the revised version.
>
> | Config. | Node Size | Skill Edge | Simi. Edge | Floor Clear | Run Score | Runtime|
> --------------|------------|-------------|-------------|---------------------|-----------------|------------------|
> | *M* = 5 | 29 | 54 | 354 | 16 | 112 | 158.34 |
> | *M* = 3 |  25 | 38 | 124 | 16 | 98 | 244.63 |
> | *M* = 10 | 22 | 34 | 116 | 11 | 63 | 192.41 |
> | RC = 1 : 1 |  29 | 54 | 354 | 16 | 112 | 158.34 |
> | RC = 1 : 2 |  33 | 32 | 220 | 16 | 102 | 153.26 |
> | RC = 2 : 1 |  20 | 26 | 114 | 13 | 87 | 145.92 |
> |  0.95, 0.88 |  29 | 54 | 354 | 16 | 112 | 158.34 |
> | 0.99, 0.90 |  155 | 190 | 14520 | 12 | 70 | 392.55 |
> | 0.85, 0.80 |  4 | 9 | 4 | 4 | 26 | 179.40 |
>
> The model exhibits robust performance across a range of configurations, with multiple hyperparameter sets yielding solid results, underscoring its general insensitivity to parameter variation, though our specific configuration delivers the best balance between task advancement and operational efficiency. We are truly grateful for this suggestion, which significantly strengthens our work.
>
> **W4. Efficiency vs. Uniformity in Module Design**
>
> **A4:** We thank the reviewer for raising this important point. While unified architectures represent a valuable future direction, our modular design was driven by critical needs for computational efficiency and operational stability in long-horizon tasks.
>
> We intentionally decouple the state encoder from the planner's VLM to maintain representation stability. The knowledge graph requires consistent state embeddings to remain coherent over extended training periods. Using the planner's dynamically-conditioned VLM embeddings would introduce instability, as identical states could receive different representations across learning stages. CLIP provides a stable semantic anchor that ensures graph consistency.
>
> While end-to-end VLM grounding is promising, our system requires frequent, pixel-accurate visual checks, particularly for "visual change" detection that gates all skill operations. For these fine-grained tasks, SAM with template matching and CLIP provide superior precision and computational efficiency compared to VLM-based localization. This approach delivers deterministic performance at lower cost (in tokens and latency), which is essential for the thousands of visual validations required during knowledge graph construction.
>
> We thank the reviewer again for this comment. We will incorporate a discussion of these design trade-offs in the revised manuscript, positioning our approach as a practical solution for current environments while acknowledging unified grounding as an important future direction.

---

> ### Author Response · Authors · 2025-11-21
> **Response (2/2)**
>
> **W2. Implementation Details**
>
> **A2:** We sincerely thank the reviewer for this critical feedback. We apologize for the lack of clarity in the original manuscript and are grateful for the opportunity to provide the following specifics. Below, we provide a point-by-point response:
>
> > What exactly is the “visual change” metric?
>
> We quantify visual change by measuring the proportion of significantly changed pixels between consecutive screenshots. The implementation follows these steps: screenshots are converted to grayscale, and the absolute difference $\Delta$ in pixel values is calculated for each corresponding position. Pixels with $\Delta$ values exceeding an intensity threshold of 30 (on a 0–255 scale) are classified as "significantly changed" and counted. The visual change ratio is computed as $\text{count} /total\_pixels$. A ratio exceeding the empirical threshold of 0.015 indicates a semantically meaningful visual transition. This approach effectively captures relevant in-game events while maintaining computational efficiency, as implemented below:
> ```
> def detect_visual_change(self, pre_screen, current_screen):
>     pre_gray = cv2.cvtColor(pre_screen, cv2.COLOR_RGB2GRAY)
>     current_gray = cv2.cvtColor(current_screen, cv2.COLOR_RGB2GRAY)
>
>     Delta = cv2.absdiff(pre_gray, current_gray)
>     _, Delta = cv2.threshold(Delta, 30, 255, cv2.THRESH_BINARY)
>     change_ratio = np.sum(Delta) / (Delta.shape[0] * Delta.shape[1] * 255)
>
>     if change_ratio > 0.015:
>         return True
>     return False
> ```
> > How are node merges decided?
>
> We determine node merges through a similarity-based clustering process that operates directly on state feature vectors. The implementation follows these steps: 1) Similarity Calculation. For each new state, we compute its feature similarity with all existing nodes using a cosine similarity metric. 2) Threshold-based Decision. If the maximum similarity exceeds our merge threshold ($\theta_{merge} = 0.85$), the state is merged with the most similar node. 3) Feature Update. Merged node features are updated to the mean of the original and new feature vectors, allowing the representation to evolve over time. 4) Graph Maintenance. Similarity edges are automatically updated to maintain the graph's connectivity structure. The key implementation is shown below:
>
> ```
> def update_knowledge_graph(self, state_feature):
>
>     max_simi = 0.0
>     most_simi_node = None
>
>     for node in exist_nodes:
>         simi = cosine(state_feature, node['feature'])
>         if simi > max_simi:
>             max_simi = simi
>             most_simi_node = node
>
>     if max_simi > 0.85:
>         node_id = most_simi_node['id']
>         ori_feature = most_simi_node['feature']
>         update_feature = mean(state_feature, ori_feature)
>         update_graph_node(node_id, update_feature)
>         update_simi_edge(node_id, update_feature)
>         return node_id
>     else:
>         new_node_id = max(node['id'] for node in exist_nodes) + 1
>         save_graph_node(new_node_id, state_feature)
>         establish_simi_edge(new_node_id, state_feature)
>         return new_node_id
> ```
> > How is edge “fitness” updated over time?
>
> The fitness of a skill edge is directly inherited from its corresponding skill. A skill's fitness $\phi(\sigma_k)$ is initially set to 0, and updated based on two factors: detectable visual state transitions following execution, and the consistency of the skill’s semantic description with the outcomes observed across before-and-after frames.
>
> > What’s the failure/termination policy?
>
> The failure/termination policy is governed by two primary conditions: 1) an upper bound of 500 steps is enforced for any single episode to maintain computational efficiency, 2) early termination is triggered by task-specific failure conditions. In Slay the Spire, this occurs when the character being defeated by the monster. In Civilization V, termination is triggered upon the nation being defeated by the AI.
>
> > What thresholds or schedules does the UCT-style selector actually use?
>
> The UCT-style selector uses C=0.5 and a base exploration threshold of 0.1, with a decaying temperature τ to transition from exploration to exploitation. It incorporates specialized mechanisms: pre-selection filtering excludes unavailable actions, a completeness penalty reduces fitness for missing prerequisites, and an exploration trigger activates when exploration probability exceeds 0.9. These are combined with numerical stabilization (max normalization, probability clipping) to ensure robust selection in partially observable environments.
>
> Thanks again for the valuable feedback. In the revised version, we have included these implementation details.
>
> In light of these responses, we hope we have addressed your concerns, and we hope you will consider raising your score. We will properly include all the rebuttal contents in the revised version, following your valuable suggestions.

---

> > ### Comment · Area_Chair_vR1A · 2025-11-26
> > **Please Review Author Response**
> >
> > Dear Reviewer,
> >
> > The authors have now responded to your comments. Could you review their response as soon as possible? If you have any further questions or concerns, please raise them as well.
> >
> > Best,
> >
> > Your AC

---

### Official Review · Reviewer_3crf · 2025-10-30

**Soundness:** 1
**Presentation:** 2
**Contribution:** 2
**Rating:** 2
**Confidence:** 3

**Summary:**

This paper introduces KG-Agent, a framework for API-free agents that aims to address inefficient exploration and short-sighted planning. The core contribution is a State-Action Knowledge Graph (SA-KG) that structures experience by linking functionally similar states (similarity edges) and successful action sequences (skill edges). This graph underpins a hybrid intrinsic reward system designed to promote strategic, long-horizon behavior.

**Strengths:**

The SA-KG is an interesting and promising data structure for agent memory. The idea of explicitly representing both state similarity and action transitions in a single graph is a notable contribution that moves beyond simple episodic memory retrieval.

The authors correctly identify two of the most critical challenges facing contemporary API-free agents: sample inefficiency ("experience siloing") and the difficulty of credit assignment for long-horizon tasks (myopia). The proposed solutions, while imperfect, are directly motivated by these well-understood problems.

Evaluating on Civilization V and Slay the Spire is commendable. These environments are significantly more challenging than typical web-browsing or application-control tasks and serve as a strong testbed for strategic reasoning.

**Weaknesses:**

While the core idea is interesting, the proposed architecture is a complex and brittle assembly of disparate components. The system stitches together a VLM for reasoning, CLIP for state embedding, SAM for UI segmentation, and template matching for action grounding, layering them beneath a custom knowledge graph, a four-part ad-hoc reward function, a two-stage skill invocation policy, and a VLM-based refinement loop. This design makes the agent fundamentally dependent on the quality of its frozen, pre-trained models and reliant on brittle heuristics, such as similarity thresholds and arbitrary reward formulas, for all its "learning" decisions. Consequently, the system is prone to unbounded growth in memory and computation, as its knowledge is simply the entire, ever-expanding graph and skill library.

The paper's central claim of developing a generalizable framework for API-free agents is not supported by its highly circumscribed experiments. The authors have selected turn-based games, a forgiving domain that conveniently masks the architecture's inherent limitations by not requiring real-time reasoning and featuring a mostly static UI with little dynamism. This narrow validation on a "home turf" benchmark does not provide credible evidence for the method's applicability to the broader, more challenging landscape of general computer interaction.

The paper's description of the skill acquisition process is not clearly explain in the paper. The proposed mechanism of incremental search over action sequences is combinatorially explosive, and the "aggressive pruning" strategy intended to manage this complexity is not defined with sufficient detail to be reproducible. Furthermore, the framework outsources the critical step of semantic abstraction to a VLM, which retroactively labels a sequence of operations rather than enabling the agent to learn their meaning internally. This issue is compounded by a significant inconsistency: the main text presents skill refinement as a generative process for creating novel behaviors, while the corresponding prompt in the appendix describes a far simpler function of de-duplicating existing skills.

**Questions:**

The knowledge graph grows with every new state and skill. Table 2 shows the number of nodes and edges increasing over rounds. What are the computational and memory implications of this growth over much longer runs (e.g., thousands of steps or dozens of episodes)? Is there a risk of the graph becoming too large to query efficiently? The authors might consider discussing potential pruning, summarization, or hierarchical abstraction strategies more in details.

The framework introduces several important hyperparameters: the merging and similarity thresholds ($θ_{merge}$, $θ_{simi}$), the reward balancing factor $\alpha$, the fitness sensitivity $C_\sigma$, and the UCT exploration constant $C$. The paper states these are held constant, which is good for reproducibility, but a brief discussion on how these were selected and how sensitive the agent's performance is to them would be valuable.

The $R_{progress}$ and $R_{semantics}$ rewards rely on a VLM's judgment. While this is a common technique, VLM evaluations can be noisy or misaligned with the true game objectives. Could the authors comment on the stability of these reward components? How much of the agent's success is dependent on the quality of these VLM-based evaluations versus the more objective graph-based rewards ($R_{state}$, $R_{novel}$)?

The paper mentions that skills are constructed through incremental increases in sequence length. For a game like Civilization V, a meaningful "skill" might span multiple turns (e.g., "move worker, build improvement"). How does the current skill definition $(\sigma = (a_1,...,a_k))$ and augmentation process handle these temporally extended, multi-turn skills? More detail on the `prompt_augment` and refinement loop would be beneficial to understand how the agent moves from atomic clicks to strategically meaningful behaviors.

The authors make a point of forgoing OCR to enhance generality. While this is a valid design choice, it could be a limitation in GUIs where critical information is presented as non-trivial small text that may be lost in the VLM input if they are resized for computation reasons. Do you have an intuition on how much the VLM is taking from its knowledge of the games and how much it is actually "observing"?

The main text (Lines 305-307) describes "skill refinement" as a generative process for creating novel skills when the agent is stuck. However, the prompt_refine provided in the Appendix (Figure 6c) is clearly designed for de-duplicating and clustering existing redundant skills. Could you please clarify the true purpose and implementation of this mechanism? Which of these two distinct functions does the agent actually perform?

---

> ### Author Response · Authors · 2025-11-21
> **Response (1/2)**
>
> We are truly thankful for your insightful and constructive review. Our detailed responses are presented below.
>
> **W1. Modularity & Scalability**
>
> **A1:** Thank you for this insightful question. KG-Agent integrates multiple components not as an ad-hoc assembly, but as a synergistic framework addressing core challenges in API-free GUI interaction: exploration inefficiency and long-horizon reasoning. Pre-trained models provide essential grounding: SAM and CLIP structure visual input for state abstraction, while the VLM handles semantic reasoning. Learning emerges from the SA-KG's topology, where state merging abstracts environment dynamics and graph-derived rewards enable extended reasoning. Scalability is maintained through state merging, which groups functionally equivalent states to constrain combinatorial growth. Planning over this compressed graph ensures tractability. This architecture demonstrates a principled balance between generality and efficiency, as evidenced by sustained performance in open-ended environments.
>
> **W2. Generalizability**
>
> **A2:** Thank you for this valuable feedback. We selected turn-based games not for their simplicity, but because they provide an ideal testbed for core challenges in API-free GUI interaction: their vast state spaces demand robust generalization, while their discrete nature cleanly separates strategic planning from real-time reactivity, allowing direct evaluation of our key contributions. While current experiments focus on near-static environments, as do other leading frameworks like CRADLE, KG-Agent's modular architecture can support dynamic settings through mechanisms like action pausing. We have clarified this scope and extension potential in the revised manuscript.
>
> **W3, Q4 & Q6. Detail of Framework**
>
> **A3:** Thank you for this valuable feedback. Following the Bottom-Up Agent paradigm, our skill augmentation is driven by successful atomic action validation rather than exhaustive search. To optimize this process, we sequentially explore three ordered atomic action subsets: starting with validated single-step skills ($A_1$), progressing to object-aware interactions ($A_2$), and finally considering background actions ($A_3$). Skill expansion terminates upon detecting any meaningful environmental feedback, at which point the functional skill is annotated and stored.
>
> For skill management, we implement dynamic pruning: First, MCTS is employed to ensure all skills receive sufficient testing opportunities. During skill execution, the agent records trajectory data and computes UCB scores according to Eq.6. Skills that exceed the average visitation count and consistently demonstrate the lowest UCB scores are pruned from the memory. As shown in Table 2, this selectively removed 1-3 skills per evaluation round, maintaining library efficiency.
>
> We leverage the VLM as a semantic abstracter to bootstrap understanding in zero-prior settings, assigning functional labels to discovered skills. Meanwhile, the SA-KG internally encodes pragmatic skill value and contextual applicability through its topology and node relations, enabling strategic generalization beyond memorized sequences. This hybrid approach allows KG-Agent to efficiently bridge low-level pixels and high-level strategy, where semantic labels support interpretability and generalization, and the graph encapsulates learned, internalized decision-making.
>
> We have revised the description of skill refinement to accurately reflect its consolidation and deduplication function, removing any misleading generative implications.

---

> ### Author Response · Authors · 2025-11-21
> **Response (2/2)**
>
> **Q1. Scalability**
>
> **A4:** Thank you for this important observation. The SA-KG's scalability is maintained through CLIP-based functional state merging, which clusters semantically similar states rather than storing unique pixel observations. As Table 2 shows, node growth saturates (26→50→53→55) while similarity edges stabilize and even decline in Round 3, confirming active graph consolidation into a compact functional model. We have added a detailed discussion on graph abstraction strategies for larger-scale deployment in the conclusion.
>
> **Q2. Hyperparameter**
>
> **A5:** Thank you for raising this point. In complex multi-agent systems, tuning a number of hyperparameters is often unavoidable. Several hyperparameters were set following established practices. For the key hyperparameters directly tied to our core innovations, we conducted a targeted sensitivity analysis in Slay the Spire (see Appendix).
>
> | Configuration | Node Size | Skill Edge | Simi. Edge | Floor Clear | Run Score | Runtime|
> --------------|------------|-------------|-------------|---------------------|-----------------|------------------|
> | *M* = 5 | 29 | 54 | 354 | 16 | 112 | 158.34 |
> | *M* = 3 |  25 | 38 | 124 | 16 | 98 | 244.63 |
> | *M* = 10 | 22 | 34 | 116 | 11 | 63 | 192.41 |
> | RC = 1 : 1 |  29 | 54 | 354 | 16 | 112 | 158.34 |
> | RC = 1 : 2 |  33 | 32 | 220 | 16 | 102 | 153.26 |
> | RC = 2 : 1 |  20 | 26 | 114 | 13 | 87 | 145.92 |
> |  0.95, 0.88 |  29 | 54 | 354 | 16 | 112 | 158.34 |
> | 0.99, 0.90 |  155 | 190 | 14520 | 12 | 70 | 392.55 |
> | 0.85, 0.80 |  4 | 9 | 4 | 4 | 26 | 179.40 |
>
> The model shows robust performance across configurations, with our chosen setup providing the optimal balance between task progress and operational efficiency.
>
> **Q3. Effect of Rewards**
>
> **A5:** Thanks for your valuable comment. We fully acknowledge this inherent limitation. To address it, we have deliberately designed a hybrid reward structure that integrates the VLM's semantic understanding of high-level strategic concepts with more objective, internally-derived game metrics. The VLM-based rewards （$R_{progress}$ and $R_{semantic}$）are specifically intended to capture hard-to-quantify strategic dimensions that cannot be easily inferred from low-level game state data. In parallel, the graph-based rewards ($R_{novel}$ and $R_{state}$) provide stable and objective signals, ensuring the agent remains grounded in fundamental game mechanics.
>
> | Setting | Library Size | Node Size | Skill Edge | Simi. Edge | Turns Survived | Techs. Unlock | Response. Rate | Token Costs |
> |------------------------------------|--------------|-----------|------------|------------|-----------------|---------------|---------------------------|--------------|
> | Full KG-Agent Model | 76 | 26 | 37  | 226| 65 | 10       | 0.89 | 3.0  |
> | w/o $R_{novel}$ & $R_{state}$ | 64 | 18 | 39  | 96         | 35  | 3 | 0.79   | 3.3          |
> | w/o $R_{progress}$ & $R_{semantic}$ | 33  | 9| 14 | 42 | 14  | 1  | 0.59 | 2.1  |
>
> Our ablation study (Table 3) confirms the critical role of VLM-driven rewards: removing both $R_{progress}$ and $R_{semantic}$ causes the most severe performance collapse, while ablating objective graph-based rewards yields more moderate declines. This demonstrates a clear synergy: graph-based rewards ensure stable exploration, while VLM rewards enable strategic advancement. The agent's success thus emerges from their collaboration within our architecture, not solely from VLM quality. We have added this analysis to the revised manuscript.
>
> **Question 5. Visual Observation or Prior Knowledge**
>
> **A5:** Thank you for raising this critical point. In fact, our design intentionally minimizes reliance on the VLM's specific knowledge of the game. This is enforced through our use of a zero-prior, task-agnostic prompt. The VLM is never informed which game it is playing and must derive all context from visual input alone. The agent's ability to learn viable policies from pixels alone demonstrates that it is performing genuine visual reasoning and not merely recalling a pre-existing game strategy.
>
> Furthermore, success fundamentally requires dynamic observation due to the combinatorial state space. The VLM's pre-trained knowledge cannot encompass near-infinite game state variations, e.g., every unique combination of resource levels, unit positions, city layouts, and technology trees. The agent's coherent sequential decision-making across this vast space confirms it interprets visual states dynamically, rather than following memorized patterns.
>
> While OCR absence may limit fine-text reading, the agent's performance under this observation-only paradigm validates that the VLM effectively "observes" to drive decisions.
>
> In light of these responses, we hope we have addressed your concerns, and we hope you will consider raising your score. We have properly included all the rebuttal contents in the revised version, following your valuable suggestions.

---

> > ### Comment · Area_Chair_vR1A · 2025-11-26
> > **Please Review Author Response**
> >
> > Dear Reviewer,
> >
> > The authors have now responded to your comments. Could you review their response as soon as possible? If you have any further questions or concerns, please raise them as well.
> >
> > Best,
> >
> > Your AC

---

> ### Author Response · Authors · 2025-12-02
> **Response Summary**
>
> We sincerely thank the reviewers for their insightful and constructive feedback. We have thoroughly addressed every question and concern raised during the discussion period. The feedback prompted us to clarify several key aspects of our methodology and experiments.
>
> In response to the reviewers' comments, we have revised the manuscript to incorporate:
> 1. Detailed explanations and empirical evidence to support our design choices.
> 2. Refined descriptions of the experimental setup and results for enhanced clarity.
> 3. New discussions that contextualize our approach within the broader landscape of AI agents.
> 4. The inclusion of two detailed case studies that further illustrate the agent's practical application.
>
> The latest version of our paper has been uploaded. For the Academic Committee’s convenience, all major modifications and additions to the text are highlighted in yellow in the document. We are confident that these comprehensive revisions fully resolve the reviewers' points and demonstrate the novelty and robustness of our proposed method.

---

### Official Review · Reviewer_yJLc · 2025-11-01

**Soundness:** 3
**Presentation:** 3
**Contribution:** 3
**Rating:** 6
**Confidence:** 4

**Summary:**

The paper builds on top of the Bottom-Up Agent framework, extending it with several new components that improve exploration and performance. It introduces a Knowledge Graph to organize experiences into a structured memory of states and actions. A hybrid intrinsic reward combines state-value estimation with novelty to balance exploration and exploitation, while enhanced visual processing (using SAM for segmentation and CLIP-like embeddings) helps the agent generalize across GUI variations. Tested on Slay the Spire and Civilization V, KG-Agent achieves better progress than Bottom-Up and other simpler baselines. The authors demonstrate how graph-structured experience representation and adaptive intrinsic rewards can make pixel-only agents more capable.

**Strengths:**

1. The motivation is strong. Tackling API-free GUI agents (pixel-only) is genuinely difficult, and the proposed SA-KG offers a clever middle ground between brute-force exploration and explicit symbolic reasoning. The method introduces a structured way to store and reuse experience, improving coherence in long-horizon behavior.
2. The *experience neighborhood* idea, linking functionally similar but visually distinct GUI states, effectively tackles a key weakness in vision-language control, where small visual shifts often break generalization. By grouping semantically equivalent states, the agent can transfer learned behaviors across different interface appearances.
3. The hybrid intrinsic reward, combining state-value estimation with novelty, is a neat and well-motivated idea that aligns with the goal of long-horizon planning. It encourages both exploration of new states and reinforcement of meaningful progress, leading to more stable and directed learning behavior.
4. The authors select the same two environments for evaluation as in the Bottom-Up paper and surpass this baseline with their improved method.
5. The SA-KG adds a layer of interpretability, allowing the agent’s exploration and decision patterns to be visualized and analyzed, unlike the more opaque skill library used in Bottom-Up Agent.
6. The ablations are quite useful, showing how the method’s core components individually contribute to performance.

**Weaknesses:**

1. **Generalizability**. I am not fully convinced how well KG-Agent could (out-of-the-box) generalizable to other application.
    a) The method relies heavily on **SAM** and **CLIP**, but both can be unreliable for GUI perception. SAM might missegment small or dynamic interface elements. For instance, segmenting nearby buttons as one region. CLIP, meanwhile, might blur fine-grained distinctions between visually similar yet functionally different states, while its feature space is directly used to infer state value and progress. The authors could investigate whether there are any failure cases caused by these components and how frequently they occur.
    b) **Hyperparameter sensitivity.** KG-Agent introduces many hyperparameters with no reported sensitivity analysis, and it is unclear whether the same configuration was used across environments. I am unsure how much manual tuning is required for each new domain. Key hyperparameters include:
        i. the coefficient balancing state-value and novelty rewards
        ii. the criterion for increasing skill length
        iii. cosine-similarity thresholds for node merging and edge creation
        iv. constants controlling fitness sensitivity and greediness weighting
        v. the number of skills sampled per execution attempt
        vi. the temperature for skill selection
    c) **Scalability**. In other environments/applications, executing a skill may require many atomic actions. As skill length grows, the MCTS-based search becomes combinatorially expensive, and SAM/CLIP must be invoked repeatedly, compounding the computational cost. Moreover, certain games naturally introduce new interface states and objects as gameplay progresses, causing the knowledge graph to also expand continuously. Without mechanisms for pruning or abstraction, the graph could grow unbounded, making retrieval and planning increasingly slow. It’s unclear how well the method scales under long-term play or whether performance degrades as the graph becomes denser and more computationally heavy to traverse.
2. The authors do not compare KG-Agent with **CRADLE**, which is odd given that CRADLE is the state-of-the-art framework for the same API-free control problem.
3. KG-Agent leans heavily on the **Bottom-Up Agent** framework but doesn’t give it nearly enough credit. A lot of the design choices, such as using SAM for segmentation, relying on implicit visual-difference rewards, the same Civilization V and Slay the Spire setup, the same evaluation metrics, and even the “zero-prior” claim, are straight out of that work. Even if the same group authored Bottom-Up Agent, that doesn’t excuse quietly absorbing its design without references. Reading the paper, it is hard to tell what is novel and what is repackaged. The authors should be more upfront about which components are inherited and what improvements their work actually contributes, rather than blurring the line between re-implementation and innovation.
4. **Evaluation**.
    a) The authors report evaluating a mere **three episodes** per environment. This leads to very high randomness in the results. After which, it is not clear whether they report results from the single best run or averages across the three runs. There is also no reporting of variance or confidence intervals, so we don’t know how consistent the KG-Agent really is.
    b) Taking **Slay the Spire** as an example, which consists of 51 floors, the KG-Agent reached the 16th, as seen from Table 1 and Figure 8. This might not be very impressive, as in the lower floors, the player only has a few different types of cards to choose from, so there are not many strategic considerations to be had. Even if playing cards at random, the player could traverse the initial 10 floors, since the game is not very punishing on the lower floors. If we were to simulate random gameplay thousands of times, it is likely that in some instances the random agent gets very lucky and reaches even beyond floor 16. Of course, it’s important to note that the random agent would not act with a keyboard and mouse, but a far simpler representation. Sure, from the results we see that the KG-Agent learns to play the game, but not how **well** it plays the game. It would be interesting to include this random agent with discrete actions (not keyboard-mouse) as an extra type of baseline.

### Typos

1. Figure 1. - Trail & Error
2. Line 472 -  proposed KG-Agent to organizes

**Questions:**

1. Is it possible to evaluate CRADLE on Civ V and Slay the Spire? Is KG-Agent better/faster/cheaper/more sample efficient than CRADLE?
2. Were the same hyperparameters used for both games? I only see the mention of “*The same agent **architecture** is deployed in both…*”.
3. When is k increased from evaluating n-step actions to (n+1)-step actions? When all combinations are exhausted?
4. How well does KG-Agent scale with skills with a long sequence of atomic actions?

---

> ### Author Response · Authors · 2025-11-21
> **Response (1/2)**
>
> We are truly grateful for the time you have taken to review our paper, your insightful comments, and support. Your positive feedback is incredibly encouraging for us! In the following response, we would like to address your major concern and provide additional clarification.
>
> **W1. a) Perception Modules**
>
> **A1:** Thank you for your valuable feedback. As with other SOTA API-free agents like Bottom-up-Agent and Cradle, our agent also builds strategic reasoning upon foundational perception modules. In response, we conducted a sampling-based analysis of its segmentation performance in Slay the Spire, finding a 98.6% success rate for operable objects. SAM reliably segments elements as small as 46×19 pixels (0.09% of image area), confirming its capability for fine-grained detection. Moreover, as shown in Fig. 5(a), KG-Agent uses CLIP to effectively distinguish visually similar screen states as separate graph nodes. We also observe a strong negative correlation between CLIP embedding similarity and pixel-wise change ratio, further validating its reliability in our visual pipeline.
>
> **W1.b) & Q2. Hyperparameters**
>
> **A2:** Thank you for this important point. While hyperparameter tuning is often unavoidable in complex multi-agent systems, we emphasize that KG-Agent uses the same hyperparameters, prompts, and experimental setups across all games, demonstrating strong generalizability with minimal manual adaptation. Several hyperparameters were set following established practices. For the key hyperparameters directly tied to our core innovations, we conducted a targeted sensitivity analysis in Slay the Spire (see Appendix).
>
> | Config. | Node Size | Skill Edge | Simi. Edge | Floor Clear | Run Score | Runtime|
> --------------|------------|-------------|-------------|---------------------|-----------------|------------------|
> | *M* = 5 | 29 | 54 | 354 | 16 | 112 | 158.34 |
> | *M* = 3 |  25 | 38 | 124 | 16 | 98 | 244.63 |
> | *M* = 10 | 22 | 34 | 116 | 11 | 63 | 192.41 |
> | RC = 1 : 1 |  29 | 54 | 354 | 16 | 112 | 158.34 |
> | RC = 1 : 2 |  33 | 32 | 220 | 16 | 102 | 153.26 |
> | RC = 2 : 1 |  20 | 26 | 114 | 13 | 87 | 145.92 |
> |  0.95, 0.88 |  29 | 54 | 354 | 16 | 112 | 158.34 |
> | 0.99, 0.90 |  155 | 190 | 14520 | 12 | 70 | 392.55 |
> | 0.85, 0.80 |  4 | 9 | 4 | 4 | 26 | 179.40 |
>
> The model shows robust performance across configurations, with our chosen setup providing the optimal balance between task progress and operational efficiency.
>
> **W1.c) Scalability**
>
> **A3:** Thank you for this point. To clarify, KG-Agent uses MCTS to select which skill to execute from a candidate set, not to choose atomic actions within skills. Thus, MCTS computational cost remains independent of skill length. To manage library growth, we implement skill refinement and evaluation for efficient maintenance. While dense knowledge graphs can impact performance, our node merging technique (validated via sensitivity analysis) mitigates this issue. As noted in the "Limitations" section, we also plan to introduce graph abstraction methods to further improve scalability.
>
> **W2 & Q1. Comparison with CRADLE**
>
> **A4:** We thank this comment. Following CRADLE's framework, we adapted its four core prompts for both games. Since CRADLE relies on predefined atomic skills, which both games do not provide, we converted KG-Agent's learned skills into CRADLE-compatible formats to ensure fair comparison.
>
> | Method | Cleared Floors | Run Scores | Execut. Rate | Token Costs | | Game Turns | Techs. Resear. | Execut. Rate | Token Costs |
> |--------------|--------------------------|-------------------|------------------------------|-----------------|-|-------------------------|---------------------|------------------------------|-----------------|
> | CRADLE       | 1.00 $\pm$ 0.00    | 5.00 $\pm$ 0.00   | NA              | 4.89 $\pm$ 0.23  | | 0.00 $\pm$ 0.00  | 0.00 $\pm$ 0.00     | NA              | 4.23 $\pm$ 0.28  |
> | Ours     | 15.67 $\pm$ 0.47 | 104.00 $\pm$ 5.89 | 0.98 $\pm$ 0.01 | 2.15 $\pm$ 0.04 | | 107.33 $\pm$ 6.55 | 13.33 $\pm$ 1.25 | 0.93 $\pm$ 0.01 | 3.03 $\pm$ 0.17 |
>
> Results show CRADLE made minimal progress in both games. This experience reinforces our motivation to develop agents that minimize manual design and prior knowledge dependence. As suggested, we have included the CRADLE comparison in the revised manuscript.
>
> **W3. Contribution**
>
> **A5:** Thank you for this observation. While KG-Agent builds upon the zero-prior, API-free paradigm established by Bottom-Up Agent, using consistent evaluation metrics and games for fair comparison, our core contribution addresses key limitations in existing API-free GUI-based frameworks: experience silos and inefficient exploration. We introduce the SA-KG to transform flat local memory into a structured, globally connected experience graph, enabling effective experience generalization and long-term reasoning. We have clarified this differentiation in the revised manuscript to better highlight KG-Agent's novel advancements.

---

> ### Author Response · Authors · 2025-11-21
> **Response (2/2)**
>
> **W4.a) Consistency of Results**
>
> **A6:** We thank this feedback. In our initial version, all methods were evaluated under the same protocol—best of three episodes—to ensure fair comparison given stochastic game initializations. Following your suggestion, we now report mean and standard deviation across three runs in the revised manuscript, providing a more statistically sound and transparent performance evaluation.
>
> | Method       | Cleared Floors | Run Scores | Execut. Rate | Token Costs | | Game Turns | Techs. Resear. | Execut. Rate | Token Costs |
> |--------------|--------------------|-----------------|-----------------|----------------|-|---------------|-------------------|-----------------|----------------|
> | GPT-4o*      | 8.33 $\pm$ 0.58    | 48.73 $\pm$ 2.43  | 0.49 $\pm$ 0.02   | 1.01 $\pm$ 0.03  | | 14.67 $\pm$ 2.08 | 1.00 $\pm$ 0.00     | 0.57 $\pm$ 0.01   | 0.91 $\pm$ 0.02  |
> | Claude3.7*   | 1.00 $\pm$ 0.00    | 5.00 $\pm$ 0.00   | 0.79 $\pm$ 0.01   | 1.22 $\pm$ 0.04  | | 19.33 $\pm$ 2.52 | 3.33 $\pm$ 0.58     | 0.92 $\pm$ 0.01   | 1.18 $\pm$ 0.09  |
> | UITARS-1.5*  | 1.00 $\pm$ 0.00    | 5.00 $\pm$ 0.00   | 0.54 $\pm$ 0.01   | 0.49 $\pm$ 0.24  | | 11.33 $\pm$ 1.15 | 1.33 $\pm$ 0.58     | 0.92 $\pm$ 0.02   | 0.12 $\pm$ 0.01  |
> | CRADLE       | 1.00 $\pm$ 0.00    | 5.00 $\pm$ 0.00   | NA              | 4.89 $\pm$ 0.23  | | 0.00 $\pm$ 0.00  | 0.00 $\pm$ 0.00     | NA              | 4.23 $\pm$ 0.28  |
> | GPT-4o       | 1.00 $\pm$ 0.00    | 5.00 $\pm$ 0.00   | 0.72 $\pm$ 0.03   | 1.27 $\pm$ 0.09  | | 6.33 $\pm$ 0.58  | 0.00 $\pm$ 0.00     | 0.37 $\pm$ 0.04   | 0.76 $\pm$ 0.03  |
> | Claude3.7    | 1.00 $\pm$ 0.00    | 5.00 $\pm$ 0.00   | 0.28 $\pm$ 0.01   | 1.08 $\pm$ 0.06  | | 0.00 $\pm$ 0.00  | 0.00 $\pm$ 0.00     | 0.15 $\pm$ 0.3    | 1.22 $\pm$ 0.13  |
> | UITARS-1.5   | 1.00 $\pm$ 0.00    | 5.00 $\pm$ 0.00   | 0.82 $\pm$ 0.01   | **0.09 $\pm$ 0.01** | | 0.00 $\pm$ 0.00  | 0.00 $\pm$ 0.00     | 0.63 $\pm$ 0.01   | **0.10 $\pm$ 0.01** |
> | BottomUp     | 14.33 $\pm$ 1.15   | 91.33 $\pm$ 9.61  | **0.98 $\pm$ 0.01** | 2.68 $\pm$ 0.29  | | 59.67 $\pm$ 8.74 | 8.66 $\pm$ 0.58     | 0.92 $\pm$ 0.01   | 3.55 $\pm$ 0.52  |
> | **Ours**     | **15.67 $\pm$ 0.47** | **104.00 $\pm$ 5.89** | **0.98 $\pm$ 0.01** | 2.15 $\pm$ 0.04 | | **107.33 $\pm$ 6.55** | **13.33 $\pm$ 1.25** | **0.93 $\pm$ 0.01** | 3.03 $\pm$ 0.17 |
>
> KG-Agent shows consistent and statistically significant superiority over all baselines across both open-ended environments.
>
> **W4.b) Random Baseline**
>
> **A7:** This perspective is interesting. In the early stages, when procedural memory is sparse and graph is unfiled, skill generation and selection are quite random. However, as the KG-Agent interacts with the environment, it learns from these random explorations, forming structured memories and improving skills within the SA-KG, enabling it to identify and reinforce effective strategies. While theoretically, purely random exploration might occasionally make progress in thousands of simulations, our experiments with a limited interaction budget show that such aimless exploration itself does not lead to sustained or meaningful progress in both games. Even powerful Claude-3.7 and GPT-4o cannot break through the first layer without prior assistance (Table 1).
>
> **Typos**
>
> **A8:** Figure 1: Trial & Reasoning, Line 472: propose KG-Agent to organize.
>
> **Q3. Skill Augmentation**
>
> **A9:** We thank this feedback. Following the Bottom-Up Agent paradigm, our skill augmentation is driven by successful atomic action validation rather than exhaustive search. To optimize this process, we sequentially explore three ordered atomic action subsets: starting with validated single-step skills ($A_1$), progressing to object-aware interactions ($A_2$), and finally considering background actions ($A_3$). Skill expansion terminates upon detecting any meaningful environmental feedback, at which point the functional skill is annotated and stored. Additional technical details have been added to the revised manuscript for clarity.
>
> **Q4: Skill Scale Concerns**
>
> **A10:** Thank you for this point. Empirically, effective skills in both game environments are typically short (k=2-3), reflecting the inherent nature of these domains where meaningful behaviors require few atomic actions. While our agent supports longer skill sequences, practical demand remains limited here.
>
> Thanks again for appreciating our work and for your constructive suggestions. We have properly included all the rebuttal contents in the revised version, following your valuable suggestions.

---

> > ### Comment · Area_Chair_vR1A · 2025-11-26
> > **Please Review Author Response**
> >
> > Dear Reviewer,
> >
> > The authors have now responded to your comments. Could you review their response as soon as possible? If you have any further questions or concerns, please raise them as well.
> >
> > Best,
> >
> > Your AC

---

### Meta-Review · Area_Chair_Fdk1 · 2026-01-07

**Summary:**

This paper presents KG-Agent, a framework that builds a knowledge graph from pixel-level GUI interactions, enabling LLM agents to explore and plan more efficiently in software without APIs.

The review scores are divergent, with two reviewers leaning slightly positive (6) and two leaning negative (4 and 2).

The key concerns are the following:
- claims of generalizability are not supported, as experiments are limited to simple, static turn-based games.
- complex and potentially and brittle architecture due to dependence on multiple frozen models and manual heuristics, resulting in unbounded memory growth. The system relies on pre-trained visual models and numerous hyperparameters, without analyzing their failures or sensitivities.
- evaluation is limited and potentially unreliable, with few episodes, no variance reporting, and a lack of comparison to simple baselines.

The rebuttal did provide additional results on consistency, comparisons with CRADLE, and sensitivity analysis. Nevertheless, the core concerns regarding the system's complexity and scalability/generalizability across different applications remain. The positive reviewers appreciate the numerous analysis of the method, but also have similar concerns regarding the use of many heuristics and modules. Without strong support from the reviewers, the AC recommends to reject.

**Reviewer Concerns:**

Reviewer yJLc: "Generalizability. I am not fully convinced how well KG-Agent could (out-of-the-box) generalizable to other application."
Rebuttal: conducted a sampling-based analysis of its segmentation performance in Slay the Spire, finding a 98.6% success rate for operable objects. This part seems to be properly addressed.

Reviewer yJLc: Missing comparison with CRADLE
Rebuttal: included the CRADLE comparison in the revised manuscript.

Reviewer yJLc:  "KG-Agent leans heavily on the Bottom-Up Agent framework but doesn’t give it nearly enough credit."
Rebuttal: clarified this differentiation in the revised manuscript to better highlight KG-Agent's novel advancements.

Reviewer 3crf: "The paper's central claim of developing a generalizable framework for API-free agents is not supported by its highly circumscribed experiments."
Rebuttal: "We selected turn-based games not for their simplicity, but because they provide an ideal testbed for core challenges in API-free GUI interaction"

This does not seem sufficiently convincing, as the claim of a generalizable framework is not properly validated by near-static environment experiments in the paper.

**Reviewer Scores:**

Reviewer yJLc: 6: marginally above the acceptance threshold.

Reviewer 3crf: 2: reject, not good enough

Reviewer V6a9: 4: marginally below the acceptance threshold

Reviewer qQ3q:  6: marginally above the acceptance threshold.

---

### Decision · Program_Chairs · 2026-01-26

Reject